# The color of melt ponds on Arctic sea ice

Peng Lu[1], Matti Leppäranta[2], Bin Cheng[3], Zhijun Li[1], Larysa Istomina[4], Georg Heygster[4]

[1]State Key Laboratory of Coastal and Offshore Engineering, Dalian University of Technology, Dalian, 116024, China
[2]Institute of Atmospheric and Earth Sciences, University of Helsinki, Helsinki, Fi-00014, Finland
[3]Finnish Meteorological Institute, Helsinki, Fi-00101, Finland
[4]Institute of Environmental Physics, University of Bremen, Bremen, 28359, Germany

*Correspondence to*: Peng Lu (lupeng@dlut.edu.cn)

**Abstract.** Pond color, which creates the visual appearance of melt ponds on Arctic sea ice in summer, is quantitatively investigated using a two-stream radiative transfer model for ponded sea ice. The upwelling irradiance from the pond surface
is determined, and then its spectrum is transformed into RGB (red, green, blue) color space using a colorimetric method. The dependence of pond color on various factors such as water and ice properties and incident solar radiation is investigated. The results reveal that increasing underlying ice thickness $H_i$ enhances both the green and blue intensities of pond color, whereas the red intensity is mostly sensitive to $H_i$ for thin ice ($H_i < 1.5$ m) and to pond depth $H_p$ for thick ice ($H_i > 1.5$ m), similar to the behavior of melt-pond albedo. The distribution of the incident solar spectrum $F_0$ with wavelength affects the pond color
rather than its intensity. The pond color changes from dark blue to brighter blue with increasing scattering in ice, and the influence of absorption in ice on pond color is limited. The pond color reproduced by the model agrees with field observations for Arctic sea ice in summer, which supports the validity of this study. More importantly, the pond color has been confirmed to contain information about meltwater and underlying ice, and therefore it can be used as an index to retrieve $H_i$ and $H_p$. Retrievals of $H_i$ for thin ice ($H_i < 1$ m) agree better with field measurements than retrievals for thick ice,
but those of $H_p$ are not good. The analysis of pond color is a new potential method to obtain thin ice thickness in summer, although more validation data and improvements to the radiative transfer model will be needed in future.

## 1 Introduction

Melt ponds are the most distinctive characteristic of the Arctic sea ice surface during summer. They can cover up to 50% of the ice surface (Webster et al., 2015) and lower the surface albedo from 0.8 (snow) to 0.15 (pond) (Perovich and Polashenski,
2012). The albedo evolution generates a positive ice-albedo feedback mechanism, which enhances the melting of ice, alters the physical and optical properties of sea ice, and even affects the salt and heat budget of the ocean surface layer (Landy et al., 2015). As a result, melt ponds play an important role in the dramatic decay of current Arctic sea ice cover (Flocco et al., 2012).

Studies on melt ponds can be categorized with respect to three aspects: morphological observations, optical measurements, and modeling of the melting processes. Morphological studies focus on the distribution and physical properties of melt ponds using field observations and remote sensing (e.g. Huang et al., 2016). The melt-pond distribution determined by aerial photography was linked to the areally averaged surface albedo (Perovich et al., 2002b), and an obvious decrease in average

surface albedo was discovered by comparing image-derived data with historical observations (Lu et al., 2010). A distinct variation trend in melt-pond fractions (MPF) in different regions of the Arctic Ocean has been found (Istomina et al, 2015) using MPF retrievals from satellite optical data (Rösel et al., 2012; Zege et al., 2015). Satellite passive microwave data were also employed to estimate MPF over high-concentration Arctic sea ice (Tanaka et al., 2016), serving as a basis for building time series of MPF in regions of consolidated ice pack. In-situ measurements of ice physics were carried out to demonstrate

the mechanisms that enable melt-pond formation (Polashenski et al., 2012), and a newly found percolation blockage process was identified to be responsible for initial meltwater retention on highly porous first-year ice (FYI) (Polashenski et al., 2017).

Optical measurements focus mainly on the partition of solar radiation in melting sea ice (e.g. Nicolaus and Katlein, 2013). The melt-pond albedo has been found to vary with the melt stage of Arctic sea ice, and the seasonal evolution of ice albedo

can be categorized into seven phases: cold snow, melting snow, pond formation, pond drainage, pond evolution, open water, and freeze-up (Perovich and Polashenski, 2012). The transmittance through FYI was almost three times larger than through multiyear ice (MYI) according to measurements made using a remotely operated vehicle under summer sea ice. It resulted from the larger melt-pond coverage of FYI compared to MYI (Nicolaus et al., 2012). Ice thickness, scattering in ice, and melt-pond distribution were found to be primary factors dominating light transmission through ponded sea ice, although their

impacts were different on small and large scales (Light et al., 2015; Katlein et al., 2015).

Finally, numerical simulations have been used to investigate the physical processes of melt ponds from formation to summertime development and then to autumn refreezing (e.g., Tsamados et al., 2015). A three-dimensional model was used to simulate the evolution of melt ponds and found that the role of snow is important mainly at the onset of melting, whereas

initial ice topography strongly controls pond size and fraction throughout the melt season (Scott and Feltham, 2010). The refreezing process of melt ponds was also modeled, and the results revealed that ice growth would be overestimated by 26% if the impact of refrozen ponds was excluded (Flocco et al., 2015). New parameterizations for melt ponds have also been embedded into climate models to evaluate the role of surface melting on the summer decay of Arctic sea ice (e.g. Holland et al., 2012). The improved models produced results that agreed more closely with observations than other models without or

only implicitly including the effect of melt ponds (Flocco et al., 2012; Hunke et al., 2013).

This study focuses on the color evolution of melt ponds on Arctic sea ice, a perspective on melt ponds that has seen few investigations so far (Perovich et al., 2002a; Light et al., 2015; Istomina et al., 2016). The photograph in Fig. 1 reveals the large variety in melt-pond appearances even on the same ice floe. The color of melt ponds varies from light bluish to dark,

largely depending on the age of the pond and the properties of the underlying ice, which can be easily examined during field investigations. First quantitative measurements of melt-pond color have been performed in the central Arctic in 2012 (Istomina et al., 2016). Beyond spectral albedo of sea ice and melt ponds measured with the portable radiometer ASD FieldSpecPro 3 (Istomina et al., 2013, 2017), a photograph has been taken at each albedo measurement site, together with ice

thickness and water depth measured by means of drilling. These field data show a clear connection between the underlying ice thickness of the melt pond and its color and spectral albedo. The effect of the water depth was found to be negligible. It has been suggested that the melt pond color can therefore be used for ice thickness estimates in summer (Istomina et al., 2016).

The motivation of this study is to elaborate on this idea and understand why the color of melt ponds can change and the micro-physical and optical reasons leading to such changes. Efforts are also made to find ways to effectively use the information provided by pond color. For example, information about sea-ice thickness below the melt pond, pond depth, and primary production in melt ponds could be retrieved.

To achieve these objectives, a radiative transfer model (RTM) initially developed to parameterize melt-pond albedo (Lu et al., 2016, hereafter LU16) is used. Section 2 introduces the color-retrieval method using the RTM. Section 3 investigates the influences of various factors, including pond depth, ice thickness, incident solar radiation, and inherent optical properties (IOPs), on melt-pond color. Section 4 discusses model uncertainty and retrievals from pond color, and Section 5 draws conclusions.

**2 Methods**

**2.1 Radiative transfer model for melt pond**

The color of a melt pond is the response of human eyes to the upwelling irradiance from the surface, which consists of the reflected solar radiation from the pond surface and the backscattering radiation from ice and water below. Based on the spectral RTM for melt ponds in LU16, each part of the upwelling radiation can be determined, thus providing the necessary

information to determine pond color.

For the two-layer model comprising of melt pond and underlying ice, radiation transfer is simplified as two streams, upwelling and downwelling irradiances. These are governed by two coupled first-order differential equations under the assumptions of diffuse incident solar radiation and isotropic scattering (Flocco et al., 2015). Assuming continuity of

radiation fluxes at air-pond, pond-ice, and ice-ocean interfaces, the irradiance in both directions in each layer can be calculated as well as the melt-pond albedo $\alpha_\lambda$ (see Eqs. (1–9) in LU16 for details).

## 2.2 Estimation of pond color from simulated upwelling spectrum

Along the whole solar spectrum, only the portion in the visible band, the wavelengths between $\lambda_1 = 380$ nm and $\lambda_2 = 780$ nm, is detectable by human eyes. To derive the color of an outgoing spectrum from the pond surface, $F_a(\lambda) = \alpha_\lambda F_0(\lambda)$ where $F_0(\lambda)$ is the incident solar irradiance, the two following methods are proposed.

The first is a mathematical method defining the color as the mean wavelength of the spectral distribution of light:

$$\bar{\lambda} = \frac{\int_{\lambda_2}^{\lambda_1} \lambda F_a(\lambda) d\lambda}{\int_{\lambda_2}^{\lambda_1} F_a(\lambda) d\lambda} , \tag{1}$$

where $\bar{\lambda}$ represents the 'mean color' of the melt pond. For example, $\bar{\lambda} = 475$ nm denotes a blue color, 510 nm green, and 570 nm yellow.

The second approach is a colorimetric method based on the fact that human eyes with normal vision have three kinds of cone cells, which sense light with spectral sensitivity peaks at long (560–580 nm), middle (530–540 nm), and short (420–440 nm) wavelengths. International Commission on Illumination (CIE, 1986) defines three color matching functions, $\bar{x}(\lambda)$, $\bar{y}(\lambda)$, and $\bar{z}(\lambda)$, as numerical description of the chromatic response of a standard observer to an incident spectrum (Fig. 2a). Note that

15 the peaks of color matching functions in Fig. 2a shift a little from those of cone cells above, and it is because modifications are necessary to avoid the mathematical difficulty as representing the color by negatives (Hunt, 2004). The tristimulus values in the XYZ color space for a reflective surface are given by:

$$\begin{cases} X = \frac{1}{N} \int_{\lambda_2}^{\lambda_1} \alpha_\lambda \cdot F_0(\lambda) \cdot \bar{x}(\lambda) d\lambda \\ Y = \frac{1}{N} \int_{\lambda_2}^{\lambda_1} \alpha_\lambda \cdot F_0(\lambda) \cdot \bar{y}(\lambda) d\lambda \\ Z = \frac{1}{N} \int_{\lambda_2}^{\lambda_1} \alpha_\lambda \cdot F_0(\lambda) \cdot \bar{z}(\lambda) d\lambda \\ \quad N = \int_{\lambda_2}^{\lambda_1} F_0(\lambda) \cdot \bar{y}(\lambda) d\lambda \end{cases} , \tag{2}$$

where $Y$ is a measure of the perceived luminosity of the light and the $X$- and $Z$- components give the chromaticity of the
20 spectrum. $N$ is defined as the reference illuminant for the reflective surface. The luminosity value ($Y$) is constrained in the range of 0–1.

The CIE XYZ color space can describe all colors visible to humans, but is not convenient for use in computer graphics or by a common output device such as an LED monitor. Therefore, the values in the XYZ space are converted into an RGB space,
25 which specifies intensity values for red, green, and blue primary light to generate a desired color. This can be done by a linear transformation as:

$$\begin{bmatrix} r \\ g \\ b \end{bmatrix} = M^{-1} \begin{bmatrix} X \\ Y \\ Z \end{bmatrix} = \begin{bmatrix} X_r & X_g & X_b \\ Y_r & Y_g & Y_b \\ Z_r & Z_g & Z_b \end{bmatrix}^{-1} \begin{bmatrix} X \\ Y \\ Z \end{bmatrix}, \qquad (3)$$

where $r$, $g$, and $b$ are the intensities of red, green, and blue primaries that yield the desired color and $M$ is the transformation matrix consisting of the coordinates of the three primaries in the XYZ space.

5    To obtain the matrix $M$, the CIE chromaticity diagram must be introduced (Fig. 2b), which describes a color in a two-dimensional chromaticity coordinate system $(x, y)$ while ignoring its luminance $Y$. The $XYZ$ tristimulus values are thus scaled as:

$$\begin{cases} x = X/(X + Y + Z) \\ y = Y/(X + Y + Z) \ . \\ z = Z/(X + Y + Z) \end{cases} \qquad (4)$$

These values are dependent, $z = 1 - x - y$, and as illustrated in Fig. 2b this two-dimensional presentation can determine the

10   given color (Hunt, 2004). For a given RGB space, the chromaticity coordinates are always given as the primary colors $(x_r, y_r)$, $(x_g, y_g)$, $(x_b, y_b)$ and the white point $(x_w, y_w)$.

According to Eq. (4), the transformation matrix $M$ can be expanded as:

$$M = \begin{bmatrix} X_r & X_g & X_b \\ Y_r & Y_g & Y_b \\ Z_r & Z_g & Z_b \end{bmatrix} = \begin{bmatrix} (X_r + Y_r + Z_r)x_r & (X_g + Y_g + Z_g)x_g & (X_b + Y_b + Z_b)x_b \\ (X_r + Y_r + Z_r)y_r & (X_g + Y_g + Z_g)y_g & (X_b + Y_b + Z_b)y_b \\ (X_r + Y_r + Z_r)z_r & (X_g + Y_g + Z_g)z_g & (X_b + Y_b + Z_b)z_b \end{bmatrix}$$

$$= \begin{bmatrix} x_r & x_g & x_b \\ y_r & y_g & y_b \\ z_r & z_g & z_b \end{bmatrix} \begin{bmatrix} X_r + Y_r + Z_r & 0 & 0 \\ 0 & X_g + Y_g + Z_g & 0 \\ 0 & 0 & X_b + Y_b + Z_b \end{bmatrix} = A \cdot S, \qquad (5)$$

where the matrix $A$ is known from Fig. 2b. To obtain the unknown diagonal matrix $S$, the definition of the white point is used. The $rgb$ intensities for the white point are $r = g = b = 1$. The luminosity is not specified in Fig. 2b; a full luminance can be used for the white point according to Eq. (2), that is, $Y_w = 1$. Substituting these values into Eq. (3):

$$\begin{bmatrix} X_w \\ Y_w \\ Z_w \end{bmatrix} = M \begin{bmatrix} 1 \\ 1 \\ 1 \end{bmatrix} \Rightarrow [X_w + Y_w + Z_w] \begin{bmatrix} x_w \\ y_w \\ z_w \end{bmatrix} = A \cdot S \cdot \begin{bmatrix} 1 \\ 1 \\ 1 \end{bmatrix}$$

$$\Rightarrow \frac{Y_w}{y_w} \begin{bmatrix} x_w \\ y_w \\ z_w \end{bmatrix} = A \cdot \begin{bmatrix} X_r + Y_r + Z_r \\ X_g + Y_g + Z_g \\ X_b + Y_b + Z_b \end{bmatrix} \Rightarrow \begin{bmatrix} X_r + Y_r + Z_r \\ X_g + Y_g + Z_g \\ X_b + Y_b + Z_b \end{bmatrix} = A^{-1} \cdot \begin{bmatrix} x_w/y_w \\ 1 \\ z_w/y_w \end{bmatrix}. \qquad (6)$$

By combining Eqs. (5) and (6), the transformation matrix $M$ is determined, and then the *rgb* intensities can be calculated using the *XYZ* tristimulus values according to Eq. (3).

Comparing the two methods, the first one is straightforward, and the result is a mean wavelength corresponding to a monochromatic light, which is not particularly good to compare with human vision or to present by computer graphics according to Fig. 2b. The second method is complex, but gives the intensity of the three primaries, so that it provides a convenient way to reproduce color on a computer. The following analyses mainly focus on the results of the latter method.

## 3 Results

To calculate radiative transfer in sea ice, certain parameters must be specified. The IOPs of sea ice and water have been fully discussed in LU16, and the results are used here. The absorption coefficients of sea ice and water ($k_{\lambda,i}$, $k_{\lambda,w}$) are shown in Fig. 3. The former is a weighted average of contributions from pure ice and brine pockets, $k_{\lambda,i} = v_{pi}k_{\lambda,pi} + v_{bp}k_{\lambda,w}$ (Perovich, 1996) and varies within $\pm 20\%$ due to varying combinations of the volume fractions of pure ice $v_{pi}$ and brine pockets $v_{bp}$ (Huang et al., 2013). The mean curve of $k_{\lambda,i}$ in Fig. 3 is defined as the absorption coefficient of Arctic sea ice in summer. Note that $k_{\lambda,w}$ is lower than $k_{\lambda,pi}$ for $\lambda < 560$ nm, and higher than $k_{\lambda,pi}$ as $\lambda > 560$ nm. The weighted average $k_{\lambda,i}$ varies closer to $k_{\lambda,pi}$ than to $k_{\lambda,w}$ because of the large volume fraction of pure ice, but sometimes it is also lower than both $k_{\lambda,pi}$ and $k_{\lambda,w}$ especially for $\lambda > 560$ nm (Fig. 3). This happens only if there are lots of gas bubbles and little brine pockets contained in sea ice, and the absorption by gas bubbles is limited but their volume fraction cannot be neglected. Scattering in meltwater and ocean water is neglected ($\sigma_{\lambda,w} = 0$). The scattering coefficient of sea ice is independent of wavelength because the scattering inhomogeneities in ice are much larger than the wavelength of light. Perovich (1990) has investigated the values of scattering coefficient for different types of snow and ice. A value of $\sigma_i = 2.5$ m$^{-1}$, corresponding to white ice interior in Perovich (1990), has been promoted by LU16 for summer Arctic sea ice because it produces more comparable melt-pond albedo with field observations than others. The value is then employed in this study. The incident solar irradiance $F_0(\lambda)$ measured by Grenfell and Perovich (2008) under a completely overcast sky on August 7, 2005 with the solar disk not visible is used because it is representative of the Arctic summer, as in LU16. The chromaticity coordinates ($x$, $y$) of the primaries are (0.640, 0.330), (0.210, 0.710), and (0.150, 0.060) for red, green, and blue respectively and (0.313, 0.329) for the white point in the selected Adobe RGB color space (Adobe, 2005). These parameters are constant throughout the study unless otherwise defined.

The color of a melt pond changes with different factors such as sky conditions, ice properties, and pond depth (Light et al., 2015; Istomina et al., 2016). We investigate the influence of various factors on pond color in the following sections.

## 3.1 Influence of pond depth and ice thickness

In this study, we assumed $H_p$ varies between 0 and 0.5 m and $H_i$ between 0.5 and 5.0 m. The range of ice thickness is somewhat beyond the current state in the Arctic summer (Lang et al., 2017). However, it is still beneficial to see the outcome of the proposed model at limiting conditions of thick deformed MYI. The results are shown in Fig. 4.

It is clear that the apparent optical properties of the melt pond are totally different for thin and thick ice. In Fig. 4a, the melt-pond albedo depends mainly on $H_i$ for thin ice ($H_i < 1.5$ m), and on $H_p$ for thick ice ($H_i > 1.5$ m), as also illustrated by LU16. The mean wavelength of pond color determined by Eq. (1) has similar features (Fig. 4b). However, the behavior of the three primary colors is somewhat different. The red intensity in Fig. 4c increases mostly with increasing $H_i$ for thin ice ($H_i < 1.5$ m), but with increasing $H_p$ for thick ice ($H_i > 1.5$ m), similarly to the wavelength-integrated albedo $\alpha_B$ in Fig. 4a. The green and blue intensities in Figs. 4d and 4e change only with $H_i$ and almost not at all with $H_p$, except for very thick ice with $H_i > 4$ m. As a result, the simulated color of the melt pond made up of the RGB components, as shown in Fig. 4f, gradually changes from dark blue to bright blue with increasing $H_i$. However, for thin ice of $H_i < 1.5$ m, the slight influence of $H_p$ on pond color is also detectable. In other words, deeper pond water makes the color bluish rather than gray because red light is more easily absorbed by pond water. Basically, melt ponds on FYI in Arctic are shallow and flat, resulting in various gray color tones, while melt ponds on MYI may have relative larger depth ranges and more complex geometrical patterns, displaying green and blue (Polashenski et al., 2012; Webster et al., 2015). These agree with the variations in Fig. 4f. The simulated pond color can be also compared with photographs during field investigations on Arctic sea ice in summer, such as in Fig. 1, which shows results that are visually close to Fig. 4f. Furthermore, the part with thinner underlying ice seems obviously darker than the rest (Fig. 1), agreeing with the trend revealed by Fig. 4f. More quantitative validations of pond color using field observations are presented in Section 3.5.

## 3.2 Influence of incident solar radiation level

Sky conditions affect the appearance of the ocean surface, but they are not considered here because of the assumption of diffuse incident radiation in the model. In this case, only the level of incident solar radiation, $F_0(\lambda)$, can be altered to investigate the influence on pond color. Except for the default value of $F_0(\lambda)$ on August 7 defined previously, five more irradiance spectra were selected according to Grenfell and Perovich (2008). All of them represent Arctic summer conditions under a completely overcast sky in August and September, 2005 (Fig. 5a). In their work, the Arctic sky was never totally clear near the solar noon in August, but in September, cloud cover decreased somewhat, providing cloud-free periods. There is also a difference in the noon solar zenith angle between August and September at 70 °N–80 °N: it is 60 °–70 ° in August and 70 °–80 ° in September. These six cases differ widely with respect to $F_0(\lambda)$. Like LU16, $H_p = 0.3$ m and $H_i = 1.0$ m are used, corresponding to a clear water pond on typical Arctic FYI, and they are constant in following discussions unless otherwise defined. The results are shown in Fig. 5b.

It is surprising that the influence of $F_0(\lambda)$ on pond color is less pronounced than that of $H_i$ and $H_p$ in Fig. 4. The $rgb$ intensities of pond color changed little under an overcast sky in August, so was the simulated color shown on the top of Fig. 5b. However, the results on overcast days in September, which produce a weaker red light but stronger blue light, show a brighter color than in August. $F_0(\lambda)$ was the only variable that could have caused the change. However, according to Fig. 5a, the incident spectra differed widely from each other and therefore were not the direct reason for the similar results in Fig. 5b.

If a normalized value of the incident irradiance is defined as $\omega = F_0(\lambda) \big/ \int_{\lambda_2}^{\lambda_1} F_0(\lambda)d\lambda$, the difference is obvious according to Fig. 6. The level of $F_0$ on an overcast day decreases with date in Fig. 5a, and $\omega$ varies with obviously stronger energy in the shortwave band (< 530 nm), but less energy in the longwave band (> 530 nm). This trend becomes more pronounced with time according to Fig. 6. As a result, the color of the melt pond in September includes more contributions from blue light, but fewer from red light (Fig. 5b).

### 3.3 Influence of optical properties of ice

Optically active inclusions in sea ice, gas bubbles, brine pockets, and biota affect the appearance and color of melt ponds on summer Arctic sea ice (Kilias et al., 2014). However, the microstructure and physical properties of sea ice cannot be treated directly by our RTM. In this section, the scattering coefficient $\sigma_i$ and the absorption coefficient $k_{\lambda,i}$, actually functions of the ice microstructure (Light et al., 2004), are investigated for their impacts on pond color (Fig. 7).

The scattering coefficient of sea ice ranges from 1.2 to 2.5 m$^{-1}$, corresponding to sea ice ranging from melting blue ice with a small content of gas bubbles to porous white ice containing large quantities of gas bubbles according to Perovich (1990). The full range starting from $\sigma_i = 0$ is presented (Fig. 7a) to understand the model outcome for an idealized purely absorbing medium. Without scattering, the melt-pond albedo is 0.05, reflecting only specular reflectance at the air-water interface, and the $rgb$ intensities of pond color are all at low level, producing a dark grey color. With $\sigma_i$ increasing into a realistic range, both the albedo and the $rgb$ intensities increase obviously, making the pond color brighter. Additionally, MYI in Arctic contains much less brine and more gas bubbles than FYI, then the more scattering in MYI is another possible factor causing the different color of melt ponds on MYI and FYI except for $H_i$ and $H_p$ (Fig. 4f).

For $k_{\lambda,i}$, the absorption coefficient of sea ice in Fig. 7b, the maximum and minimum values are determined from different combinations of volume fractions of pure ice and brine pockets (Fig. 4). With enhanced absorption in sea ice, the role of scattering in ice becomes less important, weakening the upwelling irradiance, and the albedo and the $rgb$ intensities consequently decrease. However, their changes are small compared with those shown in Fig. 7a, and the resulting variation in pond color is nearly undetectable.

The comparison in Fig. 7 clearly illustrates the importance of scattering in ice, which is the source of upwelling irradiance from the pond water and the ice interior. When scattering in ice is enhanced, the upwelling red, green and blue light from the pond surface will all be enhanced, with the red component enhanced less, producing a light blue pond color.

## 3.4 Variations during ice melt

It is interesting to see how the pond color develops during the process of ice melting. However, a complex thermodynamic model of sea ice would be needed to model in detail the changes in ice thickness and pond depth. For simplicity, an idealized model was used under the assumption of mass conservation, $H_i + \delta H_p = 1.3$ m, where $\delta$ is the ratio of water density $\rho_w$ to ice density $\rho_i$, equal to 1.3 for porous ice in summer (Huang et al., 2013). Drainage of meltwater into the ocean and basal melt of sea ice were not considered to emphasize the influence of surface melting on pond color.

During sea-ice melting, as shown in Fig. 8, the ice thickness decreases from 1.3 m to 0, and the melt pond deepens from 0 to 1 m. At the same time, the pond albedo drops from 0.5 to 0.05, and the *rgb* intensities of pond color also decrease from about 0.6 to 0.05, resulting in an evolution of the pond color from gray to blue and then to almost black.

It is also noteworthy that variations in the red band are different from those in the green and blue bands. First, the red intensity is lower overall than that of the other bands during the melting process, which can be attributed to the fact that ice and water absorb red light more thoroughly than green and blue light (Fig. 3). Second, the red intensity drops nearly linearly along with ice melt, but the green and blue intensities drop faster at the end of ice melting than at the beginning. Red decreases linearly here because it is absorbed by the growing pond, whereas green and blue can maintain higher scattering because they can penetrate the pond almost to the end.

## 3.5 Comparisons with field observations

Validation of results is important, especially for the new method presented here, but most in-situ observations of pond color are visual and qualitative. The only quantitative measurements for pond color were conducted by Istomina et al. (2016) on the Arctic sea-ice surface during the R/V Polarstern cruise ARK27/3 IceArc 2012. In addition to a portable spectroradiometer used for albedo measurements, a digital camera was used to take photographs of melt ponds, and the color information in the HSL (hue-saturation-luminance) color space was extracted to associate with concurrently measured pond depth and underlying ice thickness. The sky conditions were overcast during the optical measurements. Fog occurred frequently but its effect was limited, because the hand-held camera was close to the measured ponds and the work was stopped for heavy fog conditions. Additionally, some melt ponds observed by Istomina et al. (2016) were covered with a newly formed ice layer (1–3 cm). A new ice layer was then added to the RTM in section 2.1 to treat this situation, but the

differences between an open pond and a refrozen pond were determined to be less than 3% in the primaries of the pond color. The influence of the transparent ice layer on pond reflection is therefore ignored.

Using the measured values for $H_i$ and $H_p$, the pond color can be reproduced and compare with the in-situ observations (Fig. 9). Note that the *rgb* intensities calculated by the present model have been transformed into HSL values (0–1) to match the data in HSL color space reported by Istomina et al. (2016). The simulated pond color agrees with the in-situ measurements by Istomina et al. (2016). The correlation coefficient is $R = 0.822$ with a significance level $P < 0.01$, the root-mean-square error is $\varepsilon = 0.110$, and the average of the relative error $\langle\zeta\rangle = 37\%$. The measured $H_p$ was in the range of 8–40 cm and $H_i$ in the range of 33–256 cm, producing varying pond color with a hue value in the 0.2–0.5 range, a saturation value within 0–0.5, and a luminance value within 0.4–0.6. The correspondingly simulated hue, saturation, and luminance values of pond color were within 0.4–0.5, 0–0.3, and 0.3–0.6 respectively. Obvious divergence can be found only at individual points. For examples, points a and b in Fig. 9 belong to the same melt pond with $H_i = 0.33$ m and $H_p = 0.2$ m, but the proposed model produced a relatively large difference in the hue and luminance values of pond color compared with other points. This pond is special because it has the thinnest underlying ice layer among all the measurements. It is suspected to be a mature melt pond that will melt through to the underlying ocean, in which case the brine channels in the underlying ice layer should be much larger and denser than in other cases, with different IOPs from the present model. Point c belongs to another melt pond that has the largest saturation value among all measurements of pond color, but the proposed model reproduced a lower value.

$H_i$ and $H_p$ are variables in the calculation. Uncertainties, among others, are the different in-situ conditions from the default values in the study, such as sky conditions and ice optical properties, which need to be specified in the model since these in-situ properties were not measured in Istomina et al. (2016). We therefore carried out sensitivity studies by altering the values of $F_0$, $\sigma_i$, and $k_{\lambda,i}$ within reasonable ranges for Arctic sea ice in summer to reveal their impacts on the simulated pond color. The negative error bars in the simulated values in Fig.9 are associated with the scattering in ice as $\sigma_i$ drops from the default value (2.5 m$^{-1}$) to 1.2 m$^{-1}$ (Fig. 7a). The positive error bars are induced by $F_0$ as it decreases from the representative data in August to low levels in September (Fig. 5a). The influence of ice absorption coefficient on the simulated pond color is very limited ($< 0.02$), similar with Fig. 5b, and therefore not included in the error bars. It is revealed on Fig.9 that the impact of $\sigma_i$ on the hue and saturation values is less than 0.05, and that on the luminance is less than 0.14. On the contrary, variation in the luminance value due to $F_0$ is less than 0.04, and that in the hue and saturation is less than 0.15. That is, the maximum uncertainty in the simulated hue, saturation, and luminance values will not exceed 0.22 for different combinations of incident solar radiation and IOPs for summer sea ice. More importantly, these variations still locate almost within the range of $\pm 2\varepsilon$, namely the 95% confidence interval (Fig. 9). In other words, this experiment underlines the importance of $H_i$ and $H_p$ in determining the color of melt ponds compared with other impact factors.

## 4 Discussions

### 4.1 Uncertainties in pond-color estimation

Color is a highly subjective parameter associated with human visual perception, and therefore different people will have different descriptions even of the exact same color. Although colorimetry has provided tools to quantify and describe physically human color perception, it is still difficult to reproduce accurately the color of a reflecting surface (Fig. 9). This is true especially in the Arctic Ocean, with its severe weather conditions. Therefore, it is important to understand the limitations and uncertainties of the present method.

The first question arises from the assumption of the RTM in Section 2.1, in which diffuse incident radiation is assumed and scattering must be taken as isotropic. The former assumption is not a major problem in the summer Arctic due to the frequent presence of low stratus cloud cover. The latter assumption may, however, be inappropriate for sea ice, which possibly has more forward scattering than backward scattering, but actually most studies have still treated scattering in sea ice as isotropic (Katlein et al., 2014). Moreover, internal melting makes sea ice more porous in summer, and as a result the geometric structure of ice becomes more irregular, which can favor isotropic scattering (e.g., Leppäranta et al., 2003). Consequently, one may expect that the assumption of isotropic scattering is not much biased for melting sea ice. It is also assumed here that melt pond water is clean and scattering can be neglected (LU16). This is true if the water is meltwater from snow, and is also acceptable for ice meltwater or percolated Arctic sea water. There are no observations of any optically active impurities in melt ponds to the authors' knowledge, and the approximation has been shown valid for melt ponds shallower than 1 m (Podgorny and Grenfell, 1996). Dirty ponds with a sediment-covered floor or with cryoconite holes as observed by Eicken et al. (1994) are not considered here, and frozen melt ponds with a snow or thick ice cover in autumn (Flocco et al., 2015) are also excluded from this study.

The second question arises from the definition of the colorimetric method as retrieving the RGB components from a spectrum. Three color matching functions $\bar{x}(\lambda)$, $\bar{y}(\lambda)$, and $\bar{z}(\lambda)$, are used in Eq. (2) to quantify the chromatic response of the observer. These functions have been determined through a series of experiments that aimed to judge colors while looking through a hole with a 2° field of view (Wright, 1928; Guild, 1931). By 1960s, new color matching functions corresponding to a 10° standard observer were developed (Stiles and Birch, 1959). The 10° observer is currently believed to provide the best representation of the average spectral response of human observers, although the 2° observer still has its place for measuring objects that will be viewed at a distance. In addition, various RGB color spaces such as sRGB, Apple RGB, and Adobe RGB have been defined to satisfy the display of colors on different kinds of output devices (Süsstrunk et al., 1999), and they have different chromaticity coordinates for red, green, blue, and white colors in Fig. 3b. Tests have revealed that the differences between the two functions and among various RGB color spaces are not large enough to produce significantly different pond colors in this study, and therefore these results are not presented here.

The third question is associated with field observations of the color of melt ponds. Digital cameras used during field observations always have a viewing angle different from the standard observer defined previously, thus producing a different response to the incident spectrum. Besides, the color on photographs highly depends on the camera and the photographic parameters such as ISO and aperture values (Istomina et al., 2016), also making the direct comparisons of pond color between simulated results and field measurements difficult. Istomina et al. (2016) used RAW photographic data, which can save much more information about the light field during field observations than common image formats such as JPG, to calculate pond color. In addition, the incident solar radiation reaching the ice surface changes continuously in the Arctic Ocean, but for simplification, a constant $F_0$ was used in this study as a representative condition of the Arctic summer. However, the results shown in Fig. 5 illustrate that the influence of $F_0$ is not as important as the contributions from other impact factors.

## 4.2 Possibility of retrieving pond depth and ice thickness

Like melt-pond albedo, pond color is also affected by many factors. Among them, pond depth and underlying ice thickness are the most important according to earlier discussions. Pond color can therefore be expressed by a function such as $C = f(H_i, H_p)$ if other impact factors discussed in Section 3 are treated as empirical constants. This implies a possibility of using pond color to retrieve $H_i$ and $H_p$ through solving the inverse problem, namely $(H_i, H_p) = f^{-1}(C)$. The color $C$ is a vector comprising of red, green and blue intensities in the RGB color space or hue, saturation, and luminance values in the HSL color space.

The incident solar spectrum covers the wavelength from 300 nm to 3000 nm (Grenfell and Perovich, 2008), but most of the long waves are absorbed in the first few centimetres of water or ice because the absorption coefficients in the longwave band are larger than those in the shortwave band by at least two orders of magnitude (Warren, 1984). This means that the upwelling irradiance resulting from scattering in ice mainly consists of visible light. The color of melt ponds, which is produced by upwelling irradiance, is actually the response of the whole mass of pond water and its underlying ice regime to the incident solar spectrum, thus providing a theoretical possibility of retrieving the properties of pond water and its underlying ice from the apparent pond color.

On the other hand, the relationship between pond color and meltwater depth or sea-ice thickness has actually been qualitatively determined by many field investigations (e.g., Perovich et al., 2002a). Istomina et al. (2016) found that the underlying ice thickness has a strong impact on the saturation value of pond color, but that the effect of pond depth is small. Variations in hue and luminance values of pond color are limited and a relation to either $H_i$ or $H_p$ could not be observed. These results provided a quantitative validation of the relationship proposed here and also proved the possibility of ice property retrieval from pond color. The camera dependency of the relationship was highlighted and RAW format imagery was suggested to decrease this dependency.

Both RGB and HSL color spaces have been used in this study. Basically, they are just different mathematical descriptions of color, and without notable differences between them. The conversion between RGB and HSL is simple. The HSL color space is used to match the measurements by Istomina et al. (2016) and to examine the inverse problem $(H_i, H_p) = f^{-1} (H, S, L)$. A least-squares method is used to retrieve $H_i$ and $H_p$ from the measured pond color, and the error function is defined as the Euclidean distance between the measured and simulated pond color in the HSL color space:

$$\Delta = |(H, S, L)_{\mathrm{SIM}} - (H, S, L)_{\mathrm{MEA}}| = \sqrt{c_H \cdot (H_{\mathrm{SIM}} - H_{\mathrm{MEA}})^2 + c_s \cdot (S_{\mathrm{SIM}} - S_{\mathrm{MEA}})^2 + c_L \cdot (L_{\mathrm{SIM}} - L_{\mathrm{MEA}})^2} \,, \qquad (7)$$

where the subscript SIM denotes simulated results and MEA denotes *in-situ* measurements. The parameters $c_H$, $c_S$, and $c_L$ indicate the different sensitivity of hue, saturation, and luminance values of pond color on pond depth and ice thickness, and they are determined by normalizing the square of correlation coefficient $R^2$ between the HSL values and the measured $H_i$ and $H_p$. According to the statistical analyses in Istomina et al. (2016), there is $c_H = 0.255$, $c_S = 0.712$, and $c_L = 0.033$ (Table 1). An ergodic procedure using different combinations of $H_i$ and $H_p$ within reasonable ranges, 0.1–3 m for $H_i$ and 0.01–0.5 m for $H_p$, can be performed with an interval of 0.01 m. For each pair of $H_i$ and $H_p$, the pond color ($H_{\mathrm{SIM}}$, $S_{\mathrm{SIM}}$, $L_{\mathrm{SIM}}$) is calculated and compared with the measured color ($H_{\mathrm{MEA}}$, $S_{\mathrm{MEA}}$, $L_{\mathrm{MEA}}$), and then $\Delta$ is determined by Eq. (7). We obtain a series of $\Delta$ values through the ergodic procedure, and from the minimum $\Delta$ the retrieved $H_i$ and $H_p$ can be determined. The retrievals of $H_i$ and $H_p$ using measured HSL values by Istomina et al. (2016) are compared with measurements of $H_i$ and $H_p$ in Fig. 10.

A clear relationship between simulated and measured pond depth is not apparent (Fig. 10a), implying that the linkage between $H_p$ and melt-pond color may be somewhat loose. This result agrees with Istomina et al. (2016). The overall relationship between simulated and measured ice thickness is not clear either, but a good agreement can be found for thin ice with $H_i < 1$ m (Fig. 10b). This means, first, that the underlying ice thickness rather than the pond depth can be easily obtained from pond color, and second, that the present retrieval method is more suitable for thin ice than for thick ice.

The first statement can be partly explained by Fig. 4, which shows that the dependence of pond color on ice thickness is obviously stronger than that on pond depth except for thick ice, $H_i >1.5$ m in Fig. 4c. Moreover, the upwelling irradiance comes mainly from scattering in ice, and therefore the pond color is associated more with the underlying ice than with the pond water. The second one is associated with the assumptions in the present RTM, which treats the pond water and underlying ice as parallel layers with uniform IOPs. This assumption is more valid for thin FYI because FYI typically has larger, but shallower, ponds than MYI due to the rougher topography of MYI in general (Webster et al., 2015). Hence, measurements on MYI are more affected by the contrasts at the boundary between ponded and bare ice (Taskjelle et al., 2017), which depart from the definition of the RTM. Another possible explanation comes from ice thickness since thin ice passes through more light than thick ice. With dark ocean beneath, the thinner domain shows a better discrimination as light at some wavelengths simply does not get backscattered, and that wavelength cutoff varies quickly with ice thickness.

Nevertheless, the result shown in Fig. 10b is still encouraging. The correlation coefficient between simulated and measured ice thickness is $R = 0.819$, and the correlation is significant ($P = 0.02$). The root-mean-square error is $\varepsilon = 0.156$ m. The relative error $\xi$ presents an average of 29% and a maximum of 50%. Although the correlation coefficient for the subset of $H_i$

< 1 m in Fig. 10b is not highly increased as comparing with that in Fig. 10a, because of the few available data points in the subset, the improvements in the errors of retrievals are significant. The values of $\varepsilon$ and $<\xi>$ in Fig. 10b are approximately 1/3 and 1/2, respectively, of those in Fig. 10a. More validations from field observations are likely to improve the retrieve model in Eq. (7) and then further reduce the error in retrievals.

The results give support for a possible new method of determining the sea-ice thickness, especially for thin sea ice. Such a method would complement our knowledge about sea ice thickness since presently most sea-ice thickness retrievals from satellite remote sensing are not good during the Arctic summer because of surface melt on ice (Kwok, 2010). The limitations and applicability of the color-retrieval method are clear from the previous discussions. First, this method is valid for thin ice with thickness less than 1 m, and when the melt ponds on top of ice are open or just covered by very thin ice. Frozen melt

ponds with a snow or thick ice cover, having an obviously different appearance from open ponds, are excluded from the method. Second, overcast sky conditions are favourable for the retrieval. They are prevailing although not always present during summer in Arctic. However, further work is still needed to cover clear sky conditions. Finally, satellite remote sensing has been employed to determine MPF (e.g. Istomina et al., 2015), but it is still difficult for the satellite instruments to detect melt-pond color because the sizes of melt ponds are much smaller than the spatial resolution of most remote sensing

products. In contrast, hand-held photography (e.g. Istomina et al., 2016), ship-borne photography (e.g. Lu and Li, 2010), and airborne photography (e.g. Lu et al., 2010) are very effective ways to get the small-scale information on ice surface and provide a basis for ice thickness retrievals. For example, with unmanned aerial vehicles (UAVs) equipped with a digital camera, it is easy to observe sea ice surface features, including melt-pond color, at an ice floe scale (Wang et al., 2017).

### 5 Conclusions

A two-stream radiative transfer model was adopted and applied to ponded Arctic sea ice to examine the upwelling irradiance from the pond surface. A colorimetric method was provided to transform the upwelling spectrum into a color in the RGB color space, providing a way for comparisons with human vision and computer graphics. The dependence of pond color on the properties of the pond water and underlying sea ice was quantitatively and thoroughly investigated, and the use of pond color to retrieve the properties of ponded sea ice was also discussed.


The results reveal that both pond depth $H_p$ and underlying ice thickness $H_i$ have an important impact on pond color (Fig. 4). The green and blue intensities increase only with increasing $H_i$ except for very thick ice with $H_i > 4$ m, but the red intensity

increases mostly with increasing $H_i$ for thin ice ($H_i < 1.5$ m) and with increasing $H_p$ for thick ice ($H_i > 1.5$ m), similarly to melt-pond albedo (LU16). The reproduced pond color gradually changes from dark blue to bright blue with increasing $H_i$, visually agreeing with in-situ photography of melt ponds in the Arctic summer.

The influence of the level of incident solar irradiance, $F_0$, is limited, but its spectral distribution can cause detectable variations in pond color. The incident solar spectrum has lower radiative energy in September than in August, but it is more concentrated at short wavelengths (< 530 nm) than at long wavelengths (> 530 nm) (Figs. 5 and 6). Then the red intensity decreases, whereas the blue intensity increases as $F_0$ changes from August to September.

The IOPs of meltwater and sea ice are prescribed in the present model. In nature, the optical properties of water are more stable than those of sea ice, which change with the microstructure of ice during melting (Light et al., 2004). A sensitivity study reveals that the influence of variations in sea-ice absorption coefficient is limited, but that scattering plays an important role in pond color (Fig. 7). With increasing scattering in ice, all *rgb* intensities clearly increase, making the blue pond color brighter.

    In a simplified melt case with $H_i + \delta H_p = 1.3$ m, where $\delta = 1.3$ the ratio of water and ice density, all *rgb* intensities of pond color decrease significantly from about 0.6 to 0.05, with the resulting color varying from gray to blue and then to black. The variation in red intensity is slightly different from those of green and blue: it is lower in value, and it drops linearly with ice melt, in contrast to the nonlinear decline of the other two primary colors (Fig. 8). In a real melt process, phase transition
exists not only at ice surface but also in ice interior. If $H_i$ and $H_p$ are calculated by a thermodynamic model (e.g. Tsamados et al., 2015), and IOPs of sea ice are associated with ice physical parameters (e.g. Light et al., 2004), for example, ice porosity, then the seasonal evolutions in the color and albedo of melt ponds can be determined straightforwardly. However, it is out of the scope of the present paper and can be investigated in further studies.

The melt-pond color produced by the present model agrees with the measurements in the HSL color space reported by Istomina et al. (2016), proving the veracity of the proposed model and also implying the possibility of retrieving pond depth and ice thickness information from pond color (Fig. 9). A least-squares method was used to determine these quantities from three color components HSL. The results reveal a better agreement for ice thickness than for pond depth, and that the present model provides better retrieval for thin FYI than for thick MYI. The former is attributed to be obviously higher dependence
of pond color on ice thickness than on pond depth (Fig. 4). The latter is partly because that the plane-parallel assumption agrees more closely with ponds on flat sea ice than on rough ice, and also possibly due to the higher transparency of thin ice than thick ice.

As the first quantitative study on the color of melt ponds, this study investigated not only the extent to which pond color depends on various factors, such as $H_i$, $H_p$, $F_0$, and IOPs, but also illustrated a potential method to use pond-color data to obtain ice thickness. Many ways have been developed to obtain information on sea-ice thickness using remote-sensing technologies and drilling (Wadhams, 2005; Leppäranta, 2011), but none of them is easy and cheap to conduct in the Arctic,

and most are not feasible under summer conditions. In comparison, retrieval of ice thickness from pond color has an obvious advantage over all other methods. Hand-held, ship-borne or airborne photography of melt ponds, especially widespread UAVs equipped with a digital camera, is easy to perform during field campaigns. A recent publication by Malinka et al. (2017) suggested another way to determine pond depth and ice thickness from measured spectral albedo of melt ponds. They obtained better retrievals of $H_i$ and $H_p$ partly because they used more complicated spectra as input compared with our case.

The possibility of a color-retrieval method was explored in this study using the limited available observations so far. The authors believe that more useful information can be extracted from the color of melt ponds if further in-situ validation data can be obtained and if the RTM can be improved to suit different ice types and sky conditions.

*Acknowledgements*. This research was supported by the Global Change Research Programme of China (2015CB953901), the National Natural Science Foundation of China (41676187 and 41376186). M.L. was supported by the EU FP7 Project

EuRuCAS (European-Russian Centre for Cooperation in the Arctic and Sub-Arctic Environmental and Climate Research, Grant no. 295068) and the Academy of Finland (11409391), and B.C. was supported by the NSFC research facility mobility (41428603) and the Academy of Finland (283101). L.I. and G.H. conducted the study in the framework of the project: Spaceborne observations for detecting and forecasting sea ice cover extremes (SPICES) funded by the European Union (H2020) (Grant no. 640161). The authors are grateful to the scientific party of the ARK 27/3 cruise for making the sea ice

optical measurements possible. Special thanks are expressed to Marcel Nicolaus for organizing the logistics and to the Sea Ice Physics group on board for assisting with the measurements. Three anonymous reviewers and the editor Jennifer Hutchings are also acknowledged for their constructive comments to highly improve the manuscript.

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

**Table 1: The squared correlation coefficients $R^2$ between melt-pond color and $H_i$ and $H_p$ in Istomina et al. (2016), and the deduced coefficients $c_H$, $c_S$, and $c_L$ for Eq. (7).**

| Parameter | Coefficient | $R^2$ | | |
|---|---|---|---|---|
| | | Total | $H_i$ | $H_p$ |
| Hue | 0.255 ($c_H$) | 0.301 | 0.266 | 0.035 |
| Saturation | 0.712 ($c_S$) | 0.842 | 0.759 | 0.083 |
| Luminosity | 0.033 ($c_L$) | 0.039 | 0.020 | 0.019 |

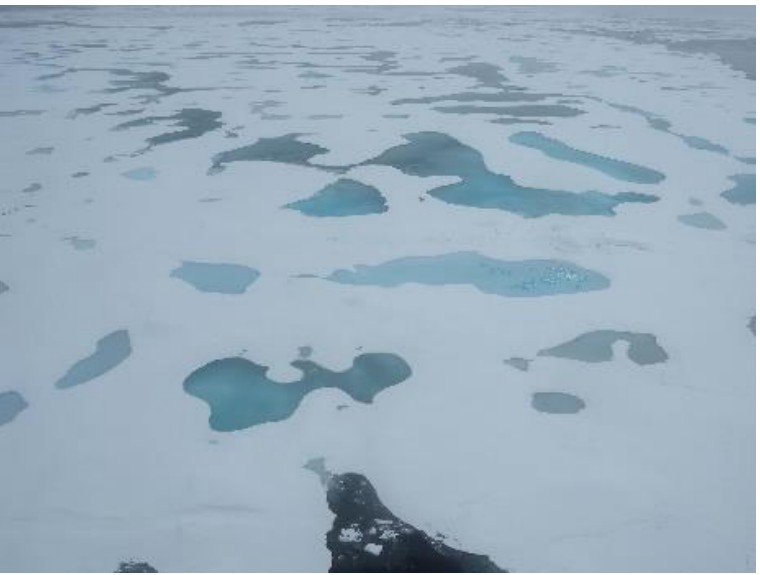

**Figure 1: A typical image of melt ponds on Arctic sea ice captured onboard R/V *Xuelong* during the Chinese National Arctic Research Expeditions in summer 2016, clearly illustrating the large variability of pond color even on the same ice floe.**

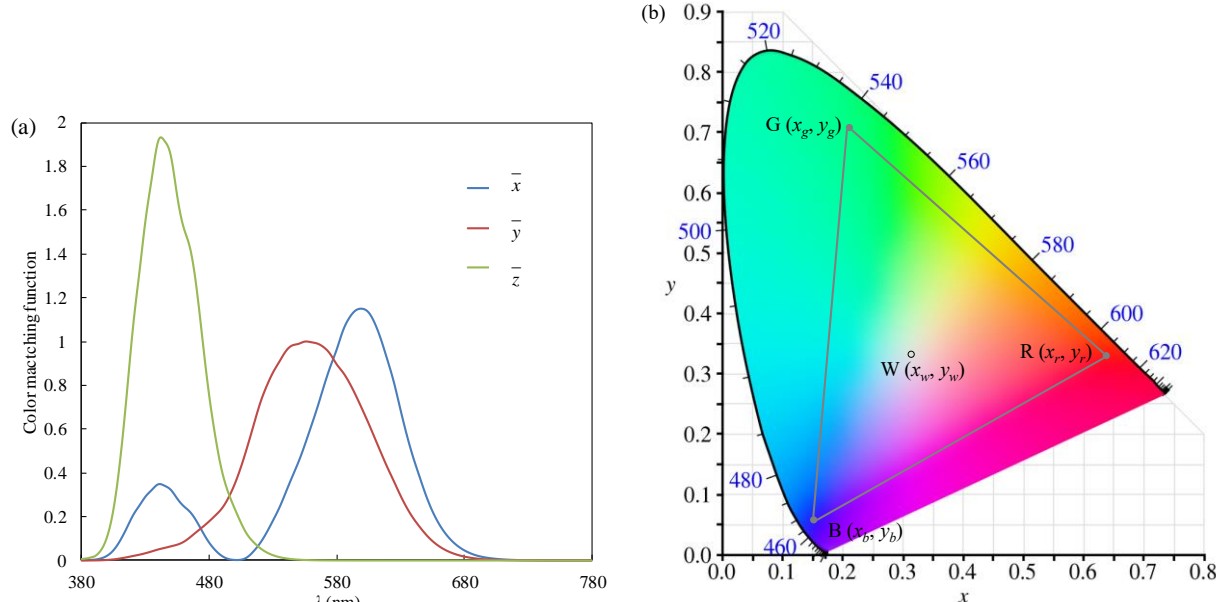

**Figure 2: (a) The CIE color matching functions $\bar{x}(\lambda)$, $\bar{y}(\lambda)$, and $\bar{z}(\lambda)$, and (b) the CIE color space chromaticity diagram. The outer curved boundary is the spectral (or monochromatic) locus, with wavelengths shown in nanometers. R, G, and B are the primary colors of red, green and blue, and W is the position of the white color.**

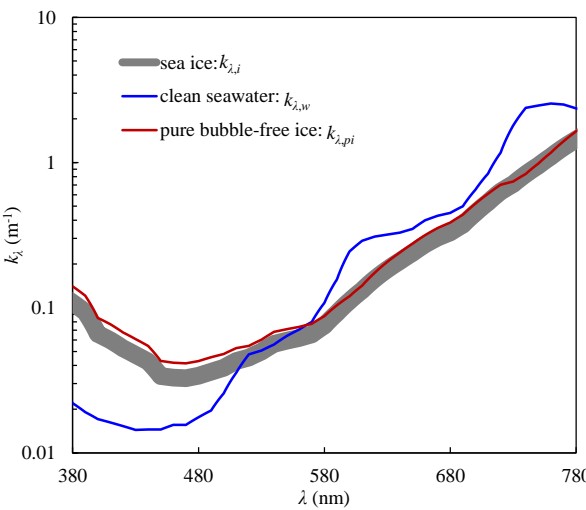

**Figure 3: Absorption coefficients of clean seawater, pure bubble-free ice and sea ice in the visible band. The water data are from Smith and Baker (1981). The pure ice data are from Grenfell and Perovich (1981) and Warren (1984). The $k_{\lambda,i}$ value was calculated from $k_{\lambda,i} = v_{pi}k_{\lambda,pi} + v_{bp}k_{\lambda,w}$, based on the volume fractions $v_{pi} \geq 60\%$ and $v_{bp} \leq 20\%$ ($v_{pi} + v_{bp} \leq 100\%$) from field observations of summer Arctic sea ice (Huang et al., 2013).**

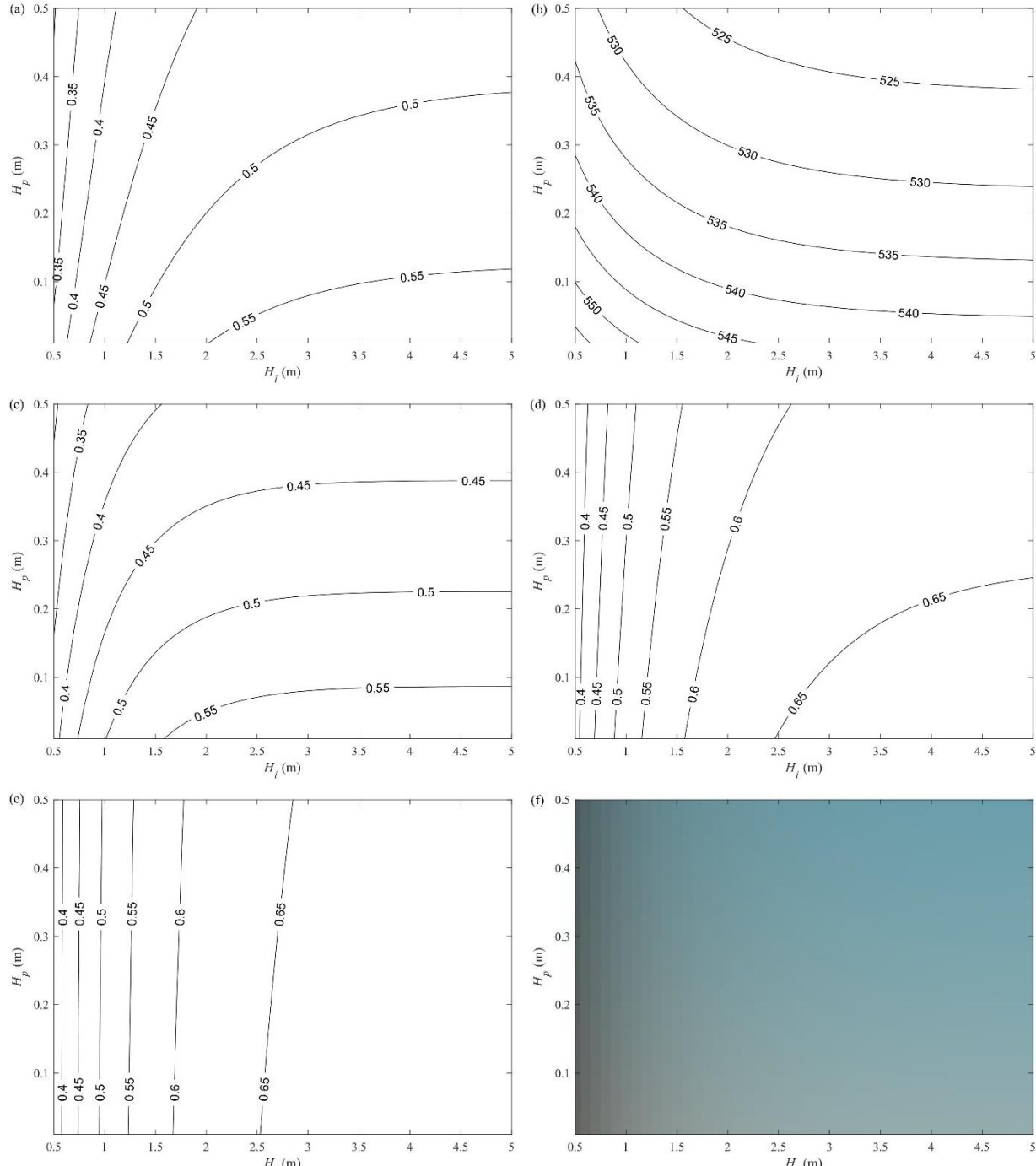

**Figure 4: Variations of melt-pond optics and color with pond depth and underlying ice thickness: (a) integrated pond albedo $\alpha_B$, (b) mean wavelength determined by Eq. (1), (c–e) intensities of red, green, and blue components scaled in the range of 0–1, (f) simulated color of the melt pond in the RBG color space according to the colorimetric method defined by Eqs. (2-6). The sky condition is overcast.**

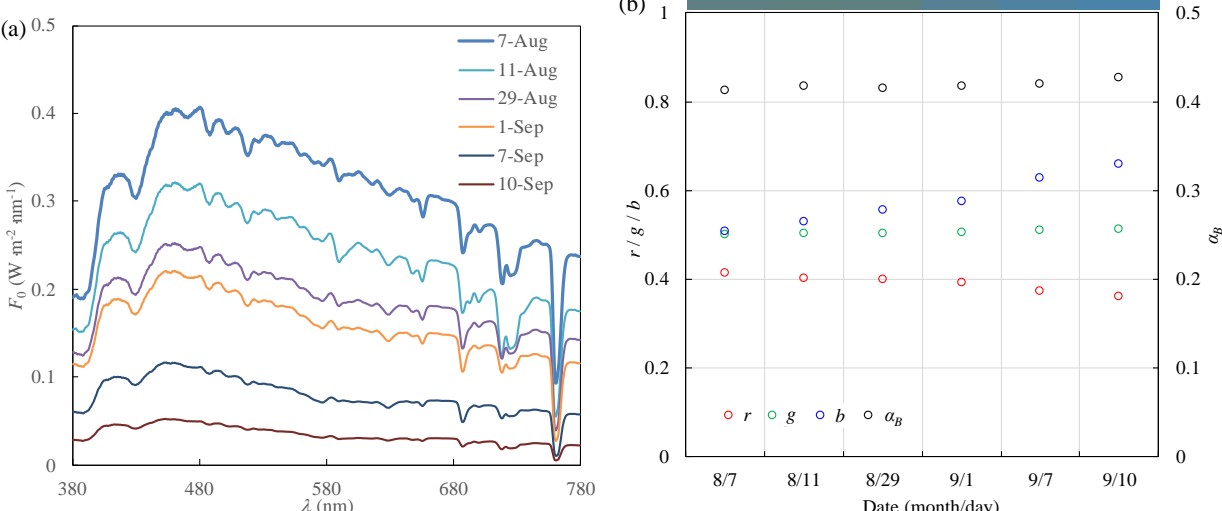

**Figure 5: (a) Typical spectral incident solar irradiances in the Arctic summer under a completely overcast sky according to Grenfell and Perovich (2008), and (b) their influence on melt-pond albedo and the *rgb* intensities of pond color for $H_p = 0.3$ m and $H_i = 1.0$ m. The color bar on top of (b) denotes the simulated color of the melt pond under different sky conditions.**

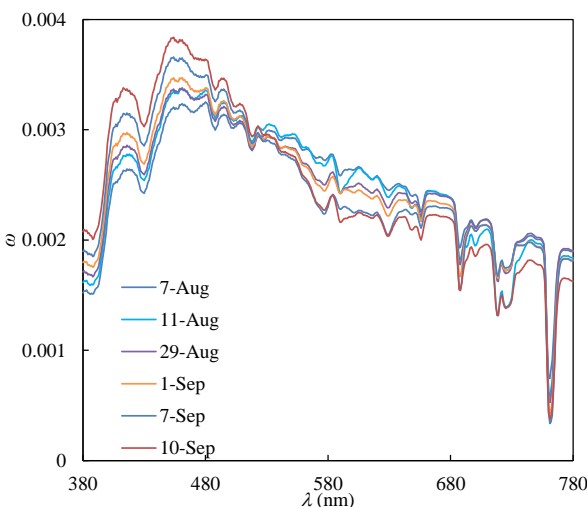

**Figure 6: Normalized values of incident solar radiation under different sky conditions, defined as the ratio of the spectrum in Fig. 5a to the total energy in the visible band.**

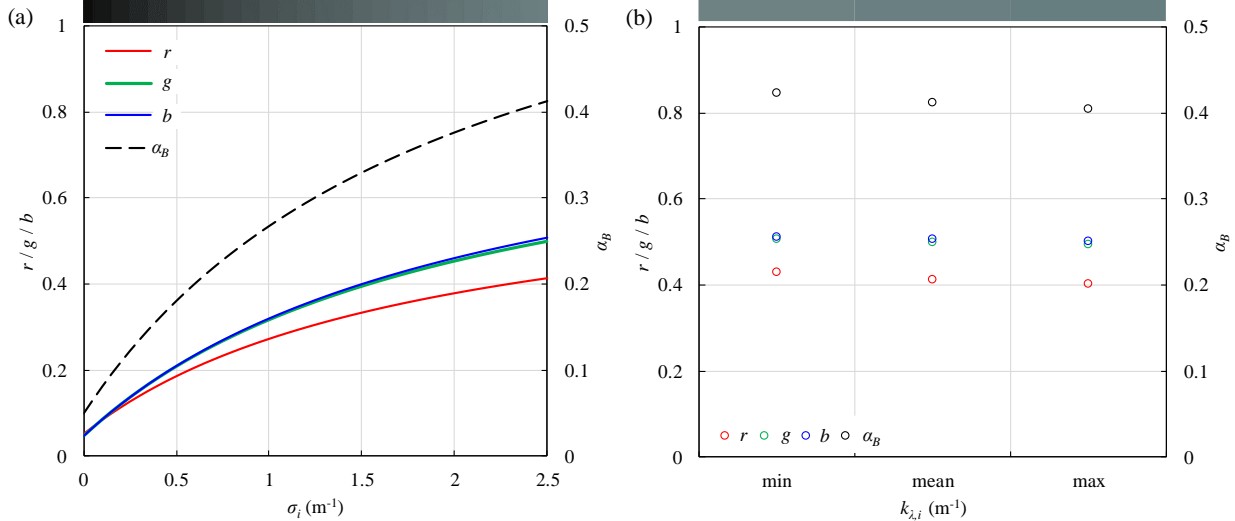

**Figure 7: Variation of the *rgb* intensities of pond color and melt-pond albedo with the inherent optical properties of underlying sea ice: (a) scattering coefficient and (b) absorption coefficient for $H_p$ = 0.3 m and $H_i$ = 1.0 m. Note that $\sigma_i$ within 1.2–2.5 m⁻¹ is valid for sea ice under melt ponds, and $\sigma_i$ = 0 is presented only as a comparison as an idealized purely absorbing medium. The color bar on top denotes the simulated color of the melt pond under different optical properties of sea ice.**

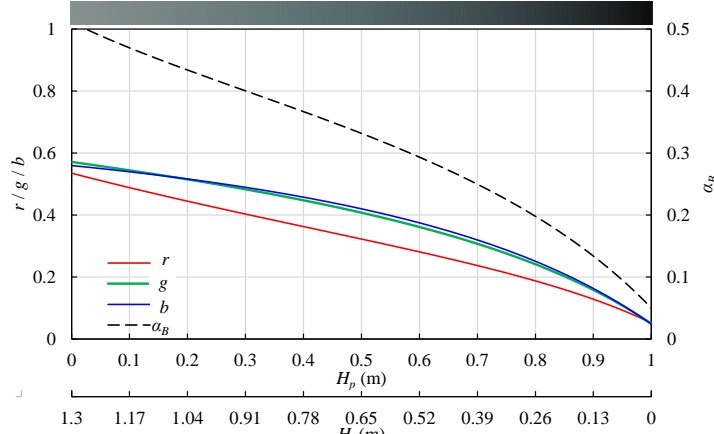

**Figure 8: Variations of the *rgb* intensities of pond color and melt-pond albedo during the process of sea-ice melting, assuming $H_i$ + $\delta H_p$ = 1.3 m. The color bar on the top denotes the simulated color of the melt pond during ice melting.**

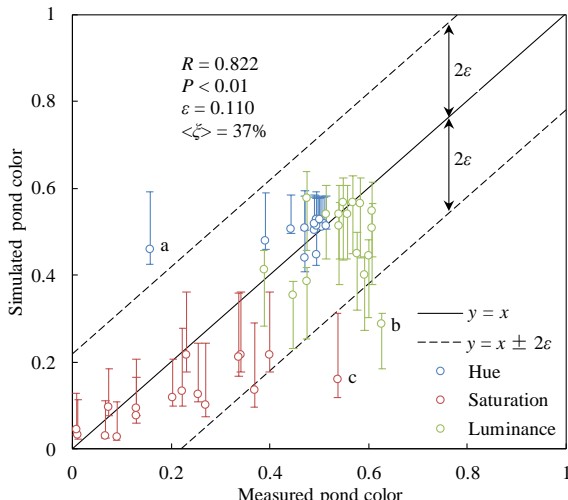

**Figure 9: Comparisons of simulated pond color with in-situ measurements by Istomina et al. (2016) in the HSL color space. Points a, b, and c are special cases discussed in the text. The vertical error bars on the simulated color denote the uncertainties due to variations in the incident solar radiation and ice scattering coefficient different from their default values. $R$ is the correlation coefficient between simulated and measured color. $P$ is the significance level of the correlation. $\varepsilon$ is the root-mean-square error, and $<\xi>$ is the mean of relative error in simulated color.**

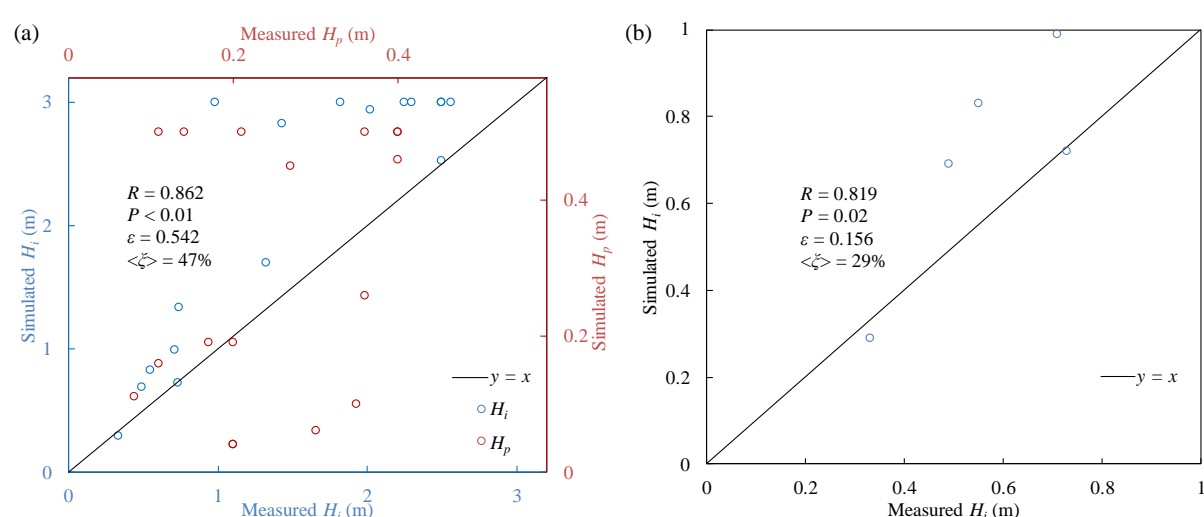

**Figure 10: (a) Retrievals of underlying ice thickness and pond depth using measured pond colors in Istomina et al. (2016). (b) is a subset of (a) for $H_i < 1$ m. $R$ is the correlation coefficient between simulated and measured $H_i$. $P$ is the significance level of the correlation. $\varepsilon$ is the root-mean-square error, and $<\xi>$ is the mean of relative error in simulated $H_i$.**