# Peer review of "The color of melt ponds on Arctic sea ice"

_The Cryosphere, 2017_

## Referee Comment (RC1) · Anonymous Referee #1 · 7 Sep 2017

This manuscript presents analysis of the spectral properties of shortwave radiation backscattered from melt ponds found on Arctic sea ice during summer. The information presented is well organized, easily readable, and clear. The subject should be of interest to readers. I have only a few minor comments, mostly technical in nature.

p. 2 line 33 photo in Fig 1 shows various evolutionary stages of ponds? not sure there is 'evolution' shown in this image. Rather, this seems to me to be a fair representation of the variety of melt pond colors often seen in a particular view, however, I see no reason to infer this field represents time-dependent changes.

p. 5 line 7 "two-dimensional representation works"– would be helpful to add a bit more information here– does the 2D representation completely describe the light field? Better to say that than 'works'.

p.10 line 12 "optically isotropic" is not the same as "isotropic scattering" line 14 same

[Figure]

question line 16 pond water 'clear with regard to its optical properties'? not at long wavelengths! p. 10 line 25 subjects' p. 14 line 2 superliner? not sure 'superliner' means p. 14 line 7 -9 I expect the reason that agreement is better for thin ice is not necessarily associated with ice topography and horizontal homogeneity assumptions of the model, but rather that thinner ice has less optical thickness. With dark ocean beneath, the thinner domain shows better discrimination as light at some wavelengths simply doesn't get backscattered, and that wavelength cutoff varies quickly with optical thickness.

Fig 2 relatively little information content here Fig 4 why does pure bubble-free ice have higher absorption than sea ice? sea water really has higher absorption than ice? These relative values surprise me, so I think they merit some comment in the text.

---

## Referee Comment (RC2) · Anonymous Referee #2 · 23 Nov 2017

The paper of "the color of melt ponds on Arctic sea ice " give a new insight on the optical properties of Arctic melt pond, which is very important for the knowledge of melt pond thermodynamic processes and remote sensing. There are very few papers have been published on this topic because of the complexity of influencing factors. Thus, it is worthy of publication. However, in the current state, I think this study just can give the knowledge on the color of idealized and simple melt pond because it just give the model of two-layer pond (ice covered by water) and just in the state of overcast sky: (1) Most melt ponds in Arctic would be covered by a thin ice although in the midsummer because the cold air at night, and the snow accumulated on the thin ice and itself would influence the optical characteristics of melt pond, as shown in the Fig.1 ( Not all of them are open melt pond) ; (2) overcast sky is prevailing but not always during summer in Arctic and the incident spectrum would obvious influence the pond color. Thus, if the authors can add some works on (1) three- or four- layers model and (2)

the influences of incident spectrum, this study would be effectively improved both for the preciseness and applicability. Here are some other detail comments: (1) Color can be equivalent to albedo. Color only covers the visible light. (2) Scattering in meltwater and ocean water is neglected. Whyïij§ (3) 3.3 Influence of optical properties of ice. – how about the porosity of the ice under the melt water. Many cases, the density of ice under melt pond is only about 1/3 of that of level ice because the large porosity and the salinity is as large as the upper ocean. (4) 4.2 Possibility of retrieving pond depth and ice thickness.—I would like remove this section because: (1) the visible color of pond is very difficult to obtained by satellite remote sensing due to the cloud and small scale of the pond, (2) we cannot judge which pond is covered by ice and/or snow by satellite/aerial images, (3) the color of pond also depends on many factors, especially for the porosity of ice under the ice, it also can be found that the relationship is very unreliable as shown in fig.11.

Please also note the supplement to this comment:
https://www.the-cryosphere-discuss.net/tc-2017-117/tc-2017-117-RC2-supplement.pdf

---

## Editor Comment (EC1) · J. Hutchings (Editor) · 27 Nov 2017

Dear Peng Lu and co-authors,

Thank you for your contribution. I am interested in receiving your response to the reviews. I have provided some additional comments below. At this stage I have not checked the paper for continuity. I do note that the paper is well written and English clear (thank you), and as I expect you will make some substantial revisions to the paper I am holding off on a through proof-read until after revision.

Please consider another paper in the Cryosphere Discussion that is on the topic of reflectance of melt ponds. I would be very interested in your opinion on the complimentary nature of your work to this. You can find the paper at https://www.the-cryosphere-discuss.net/tc-2017-150/tc-2017-150.pdf, or I can send you a pdf if you need.

[Figure]

In general, please check that you are citing the original source of information. Was Polashenski and Perovich (2012), line 15, page 2, the original source of the 7 stage model for albedo evolution in summer? I recall Hajo Eicken and Don Perovich talking about this much earlier.

I am curious, could your model be extended to clear skies with non-diffuse illumination? Would this allow you to identify the thickness of ice under melt ponds from satellite imagery such as provided by MODIS? Is this inverse problem one you considered? How much influence does assuming overcast skies have on the comparison with in-situ observations. Did you only consider the sub-set of overcast data in the comparison, or does this include data for all skies?

In the figure captions, I assume "true color" refers to the modeled color of the melt pond. Can you clarify.

I had some difficulty in following your discussion on retrieval of ice thickness from pond color. I feel you need to clarify the discussion as to the parameters that confound the inverse solution. It would help to provide the evidence for this. In particular the paragraph on lines 10-15, page 12, is vague what the competing parameters in the sky and ice conditions are and how they counteract each other such that it might not be possible to find a single solution based on meltpond color. Given you are justifying the value of your work based on the possibility of developing ice thickness products from satellite and camera observations I feel this needs to be addressed much more carefully in your analysis and discussion.

Please consider acknowledging those who collected the data you use in this study.

Looking forward to your response, Jenny

---

## Referee Comment (RC3) · Anonymous Referee #3 · 29 Nov 2017

General Comment:

This paper describes investigation of the color of melt pond on the Arctic sea ice simulated by the radiative transfer model and validation using field observations. Such sensing and analyzing melt pond may become increasingly important for detecting progress of warming in the Arctic Ocean. Therefore I recommend this paper for publication. However, I have a couple of major and minor comments that should be considered.

Major comment 1): In section 3.1, the authors mention the effects of melt depth and underlying ice thickness (P6. L18 – P7.L5). In addition, the effects of albedo and color of melt pond are also considered. The authors describe the pond color depends on underlying ice thickness and the possibility of estimation of ice thickness from the pond color. Basically, the pond color on first-year ice (FYI) containing brine and sea water indicates various gray depending on pond depth. The pond color on multi-year

ice (MYI) displays green and blue. Thus the pond color also depends on underlying ice types (FYI or MYI). I recommend to add a description about the effect of ice type difference for same ice thickness. This explanation is expected to make the validity of this manuscript increase.

Major comment 2): I agree the result of the comparisons with field observations described in section 3.5. However, the description of the quantitative measurements for pond color by Istomina et al. (2016) is incomplete. Fog appears frequently during summer and observation of pond color seem to be affected by fog. The authors should mention the influence of fog during summer.

Major comment 3): According to Fig. 11, a good agreement can be found for thin ice with ice thickness < 1 m (P12. L19-L20). I would like to suggest that the color-retrieval method using a RTM is useful to estimate thin ice thickness because sea ice thickness has been declined in recent years. This is not discussed in a convincing way. In order to understand the argumentations given in the manuscript, I recommend to add discussion about when and where the color-retrieval method is useful. I think the valid area and period of the color-retrieval method are mainly ice edge and in late-summer, respectively.

Major comment 4): The manuscript describes that the result shown in Fig. 11 is still encouraging (P13. L1-L5). However, it is difficult to agree a new way of determining the sea-ice thickness. To clarity the validity of the color-retrieval method using RMT, I recommend to redraw plots of the ice thickness less than 1 m and more than 1 m separately in Fig. 11b. Adding the correlation coefficients, significance levels, and root mean square errors in Fig. 11 is also recommended.

Minor comments:

1) P2-L1: Studies on melt ponds area more than three aspects. For example, the studies using synthetic aperture radar and passive microwave sensor should be included. There are not many papers about remote sensing of melt pond by satellite. Recently
Tanaka et al. (2016) reported estimation of melt pond fraction using satellite microwave radiometer. I recommend to cite their paper in this section. Tanaka, Y., K. Tateyama, T. Kameda, and J. K. Hutchings (2016), Estimation of melt pond fraction over high con­centration Arctic sea ice using AMSR-E passive microwave data, J. Geophys. Res. Oceans, 121, doi:10.1002/2016JC011876.

3) P3-L14: RTM was investigated the dependence of apparent optical properties (AOPs), particularly albedo and transmittance, on sky conditions, pond depth, ice thick­ness, and the inherent optical properties (IOPs) of ice and water (Lu et al., 2016). That is worth mentioning as well. For example, it would be essential to show about the broadband albedo were higher on overcast days than on clear days by 0.01 in August.

---

## Author Comment (AC1) · 20 Dec 2017

The authors are grateful to the referee for the constructive comments and additional grammar corrections. They have been helpful to improve the manuscript significantly. Our responses are addressed point to point hereafter.

This manuscript presents analysis of the spectral properties of shortwave radiation backscattered from melt ponds found on Arctic sea ice during summer. The information presented is well organized, easily readable, and clear. The subject should be of interest to readers. I have only a few minor comments, mostly technical in nature.

p. 2 line 33 photo in Fig 1 shows various evolutionary stages of ponds? not sure there is 'evolution' shown in this image. Rather, this seems to me to be a fair representation of the variety of melt pond colors often seen in a particular view, however, I see no reason to infer this field represents time-dependent changes.
Reply: Indeed, the "evolutionary" is not very accurate. We revised this sentence to "The photograph in Fig. 1 reveals the large variety in melt-pond appearances even on the same ice floe."

p. 5 line 7 "two-dimensional representation works"– would be helpful to add a bit more information here– does the 2D representation completely describe the light field? Better to say that than 'works'.
Reply: Corrected accordingly. We revised this sentence to "These coordinates are dependent, z = 1 - x - y, and as illustrated in Fig. 2b this two-dimensional presentation can determine the given color (Hunt, 2004)".

p.10 line 12 "optically isotropic" is not the same as "isotropic scattering" line 14 same question line 16 pond water 'clear with regard to its optical properties'? not at long wavelengths!
Reply: We revised the terms in lines 12 and 14 into "isotropic scattering". It is exactly what we want to express there. The sentence in line 16 was revised to "it is assumed here that melt pond water is clean and scattering can be neglected (LU16)".

p. 10 line 25 subjects'
Reply: Corrected accordingly. The sentence was revised to "These functions have been determined through a series of experiments that aimed to judge colors while looking through a hole with a 2° field of view".

p. 14 line 2 superliner? not sure 'superliner' means
Reply: The description was not correct in the original manuscript. It should be "nonlinear".

p. 14 line 7 -9 I expect the reason that agreement is better for thin ice is not necessarily associated with ice topography and horizontal homogeneity assumptions of the model, but rather that thinner ice has less optical thickness. With dark ocean beneath, the thinner domain shows better discrimination as light at some wavelengths simply doesn't get backscattered,

and that wavelength cutoff varies quickly with optical thickness.

Reply: We agree with reviewer. We added such interpretation in the revised manuscript at P13 "Another possible explanation comes from ice thickness since thin ice passes through more light than thick ice. With dark ocean beneath, the thinner domain shows a better discrimination as light at some wavelengths simply does not get backscattered, and that wavelength cutoff varies quickly with ice thickness" as explaining the figure.

At P15 "The latter is partly because that the plane-parallel assumption agrees more closely with ponds on flat sea ice than on rough ice, and also possibly due to the higher transparency of thin ice than thick ice" in the conclusion.

Fig 2 relatively little information content here

Reply: The original Figure 2 (Schematic graph of the radiative transfer model) was removed accordingly. The model was explained by text in P3.

Fig 4 why does pure bubble-free ice have higher absorption than sea ice? Sea water really has higher absorption than ice? These relative values surprise me, so I think they merit some comment in the text.

Reply: The absorption coefficients employed in the study came from different references. Data of the absorption coefficient of water came from Smith and Baker (1981). Data of the absorption coefficient of pure ice came from Grenfell and Perovich (1981) (> 400 nm) and Warren (1984) (< 400 nm). According to these data, the absorption coefficient of water is a little higher than that of pure ice in the 560-780 nm band, but lower than that of pure ice in the 380-560 nm band.

The absorption coefficient of sea ice is a weighted-average of that of water and pure ice according to Perovich (1996), its values are closer to pure ice than to seawater because of the large volume fraction of pure ice. Sometimes, the absorption coefficient of sea ice is also lower than that of pure ice and seawater, especially as wavelength greater than 560 nm in the figure. This happens only if there are lots of gas bubbles and little brine pockets contained in sea ice, and the absorption by gas bubbles can be ignored but their volume fraction cannot be neglected.

We added the new references into the figure capital as "Figure 3: Absorption coefficients of clean seawater, pure bubble-free ice and sea ice in the visible band. The water data are from Smith and Baker (1981). The pure ice data are from Grenfell and Perovich (1981) and Warren (1984). The $k_{\lambda,i}$ value was calculated from $k_{\lambda,i} = \nu_{pi}k_{\lambda,pi} + \nu_{bp}k_{\lambda,w}$, based on the volume fractions $\nu_{pi} \geqslant 60\%$ and $\nu_{bp} \leqslant 20\%$ ($\nu_{pi} + \nu_{bp} \leqslant 100\%$) from field observations of summer Arctic sea ice (Huang et al., 2013)".

And we also added comments in P6 as "Note that $k_{\lambda,w}$ is lower than $k_{\lambda,pi}$ for $\lambda < 560$ nm, and higher than $k_{\lambda,pi}$ as $\lambda > 560$ nm. The weighted average $k_{\lambda,i}$ varies closer to $k_{\lambda,pi}$ than to $k_{\lambda,w}$ because of the large volume fraction of pure ice, but sometimes it is also lower than both $k_{\lambda,pi}$ and $k_{\lambda,w}$ especially for $\lambda > 560$ nm (Fig. 3). This happens only if there are lots of gas bubbles and little brine pockets contained in sea ice, and the absorption by gas bubbles is limited but their volume fraction cannot be neglected".

References

Grenfell, T.C., and D.K. Perovich. 1981. Radiation absorption coefficients of polycrystalline ice from 400–1400 nm. Journal of Geophysical Research, 86: 7447–7450.

Hunt R.G.W. 2004. The reproduction of colour, 6th ed. John Wiley & Sons, pp. 844.

Perovich, D. K. 1996. The optical properties of sea-ice. Cold Reg. Res. and Eng. Lab. (CRREL) Report 96-1, 585 Hanover, NH.

Smith, R.C., and K.S. Baker. 1981. Optical properties of the clearest natural waters (200–800 nm). Applied Optics, 20: 177–184.

Warren, S.G. 1984. Optical constants of ice from the ultraviolet to the microwave. Applied Optics, 23, 1206-1225.

---

## Author Comment (AC2) · 20 Dec 2017

We thank the referee for the comments on our manuscript. The comments have been helpful to improve the manuscript a lot. Our responses are addressed point by point below.

The paper of "the color of melt ponds on Arctic sea ice" give a new insight on the optical properties of Arctic melt pond, which is very important for the knowledge of melt pond thermodynamic processes and remote sensing. There are very few papers have been published on this topic because of the complexity of influencing factors. Thus, it is worthy of publication. However, in the current state, I think this study just can give the knowledge on the color of idealized and simple melt pond because it just give the model of two-layer pond (ice covered by water) and just in the state of overcast sky: (1) Most melt ponds in Arctic would be covered by a thin ice although in the midsummer because the cold air at night, and the snow accumulated on the thin ice and itself would influence the optical characteristics of melt pond, as shown in the Fig.1 ( Not all of them are open melt pond) ; (2) overcast sky is prevailing but not always during summer in Arctic and the incident spectrum would obvious influence the pond color. Thus, if the authors can add some works on (1) three- or four- layers model and (2) the influences of incident spectrum, this study would be effectively improved both for the preciseness and applicability.

Reply: (1) We investigated the case that a thin-ice layer is placed on top of the melt pond (three-layer model) in section 3.5 as comparing simulated color with field observations, as some observed melt ponds by Istomina et al. (2016) were indeed covered by a very thin ice layer as the reviewer said. However, the differences in the results determined by an open pond model and an ice-covered pond model are very limited, and less that 3% in the HSL values of the pond color (as shown on Figure A below). That means the influence from the transparent ice layer (1–3 cm) on pond reflection can be ignored.

[Figure]

Figure A. Comparison between the simulated color of an open pond and a frozen pond. Note that this figure is only used to show here, and will not be included into the revised manuscript because the comparison in the figure is straightforward and can be explained clearly by text.

(2) The incident solar spectrum is different day to day although under overcast sky conditions. In section 3.2, we selected six different measurements of $F_0$ according to Grenfell and Perovich (2008), and then investigated the influence of solar spectrum on the color of melt pond. A diffuse incident solar radiation is the basic assumption of the present twostream radiative transfer model, so the influence of the direction of solar beam in clear days cannot be investigated in this study. It is the same with the studies in Perovich (1990), Taylor and Feltham (2004), Flocco et al. (2015) who employed the similar two-stream radiative transfer model. Additionally, as the reviewer said, overcast sky is prevailing but not always during summer in Arctic. It is acceptable if most, not all, situations can be treated in a single paper. Of course, we agreed that "further work is still needed to cover clear sky conditions." (in section 4.2 and conclusions).

Here are some other detail comments:
(1) Color can be equivalent to albedo. Color only covers the visible light.
Reply: We agree. Both color and albedo are representations on the spectral radiation reflected back from the pond surface. The differences between them are (1) color covers only the visible band, but albedo covers a larger band, for example, 350 – 950 nm, if measured by a RAMSES radiometer; (2) color can be sensed by CCD cameras or human eyes, but albedo can only be measured by a radiometer. Color is more easily to be measured and observed directly by human eyes, so a study on the color of melt pond is necessary although extensive studies have been conducted on the albedo of melt pond.

(2) Scattering in meltwater and ocean water is neglected. Why?
Reply: We ignored the scattering in water because (1) this has been shown to be a valid approximation for melt ponds with a depth less than 1 m (Podgorny and Grenfell, 1996a; Taylor and Feltham, 2004). (2) The scattering coefficient of pure water is 2-3 orders of magnitude lower than that of sea ice (Smith and Baker, 1981), scattering in water is therefore not a main factor affecting the optics of melt pond as comparing with ice scattering. (3) There are no observations of any optically active impurities in melt ponds to the authors' knowledge. (4) Clear melt ponds are the focus of this study, and dirty ponds with a sediment-covered floor or with cryoconite holes as observed by Eicken et al. (1994) have been excluded. (5) The ocean beneath ice is always regarded as a semi-infinite medium and there is no radiation scattered upward within the ocean, for examples, in Taylor and Feltham (2004), and Lu et al. (2016). As a result, no scattering is an acceptable approximation for meltwater and ocean.
    We added these explanations in section 4.1.

(3) 3.3 Influence of optical properties of ice. – how about the porosity of the ice under the melt water. Many cases, the density of ice under melt pond is only about 1/3 of that of level ice because the large porosity and the salinity is as large as the upper ocean.
Reply: Ice porosity is indeed an important parameter of melting sea ice. However, it cannot be directly included into the radiative transfer model in this study. Instead, the influence of ice porosity was investigated through considering ice absorption and scattering coefficients. In Fig. 7a, different values of ice scattering coefficient corresponded to different content of gas bubbles in sea ice, which has been studied in Perovich (1990). In Fig. 7b, the absorption coefficient of sea ice was calculated by the weighted-average of that of water and pure ice, and the ice absorption coefficient is actually determined by the volume fractions of pure ice and brine pockets in sea ice. As a result, although ice porosity is not explicitly included in section 3.3, it poses an impact on both absorption and scattering in sea ice, and further on

the color of melt pond.

We clearly stated these now in section 3.3 as "However, the microstructure and physical properties of sea ice cannot be treated directly by our RTM. In this section, the scattering coefficient $\sigma_i$ and the absorption coefficient $k_{\lambda,i}$, actually functions of the ice microstructure (Light et al., 2004), are investigated for their impact on pond color".

Also we presented a possibility to fully consider ice porosity in the conclusion section: "In a real melt process, phase transition exists not only at ice surface but also in ice interior. If $H_i$ and $H_p$ are calculated by a thermodynamic model (e.g. Tsamados et a., 2015), and IOPs of sea ice are associated with ice physical parameters (e.g. Light et al., 2004), for example, ice porosity, then the seasonal evolutions in the color and albedo of melt ponds can be determined straightforwardly. However, it is out of the scope of the present paper and can be investigated in further studies".

(4) 4.2 Possibility of retrieving pond depth and ice thickness.—I would like remove this section because: (1) the visible color of pond is very difficult to obtained by satellite remote sensing due to the cloud and small scale of the pond, (2) we cannot judge which pond is covered by ice and/or snow by satellite/aerial images, (3) the color of pond also depends on many factors, especially for the porosity of ice under the ice, it also can be found that the relationship is very unreliable as shown in fig.11.

Reply: We disagree with reviewer on this comment and would still prefer to keep our discussion in this section because:

(1) We were not promoting retrieve ice thickness from melt pond colors that are detected by the satellite data. Instead, we would argue that "hand-held photography, ship-borne photography, and airborne photography are very effective ways to get the small-scale information on ice surface" and to be used to retrieve thin ice thickness. Especially, with the wide applications of unmanned aerial vehicles (UAVs) in sea ice investigations, it is easy for UAVs equipped with a digital camera to get the information of pond color.

(2) It is indeed difficult to judge if the pond is covered by ice or snow by images. However, according to our newly added analyses in section 3.5 and Figure A above, a thin ice cover on top of a melt pond does not change the color of the melt pond very much. So the error introduced by the thin ice cover can be ignore as retrieving ice thickness from pond color. The ponds that covered by snow or thick ice are most likely beyond the Arctic summer season and are therefore excluded from this study.

We have added a new paragraph in section 4.2 to clarify the limitations and applicability of the color-retrieval method, including the text presented in (1) and (2).

(3) The color of melt pond indeed depends on many factors, as we have investigated in sections 3.1, 3.2, and 3.3. However, once we identified the primary factors, the pond color can be determined. And We have improved the retrieve model (c.f. Eq. 7) and the results showed some improvement for thin ice thickness detection:

$$\Delta = |(H,S,L)_{\mathrm{SIM}} - (H,S,L)_{\mathrm{MEA}}| =$$

$$\sqrt{c_H \cdot (H_{\mathrm{SIM}} - H_{\mathrm{MEA}})^2 + c_S \cdot (S_{\mathrm{SIM}} - S_{\mathrm{MEA}})^2 + c_L \cdot (L_{\mathrm{SIM}} - L_{\mathrm{MEA}})^2} \ , \qquad (7)$$

The parameters $c_H$, $c_S$, and $c_L$ indicate the different sensitivity of hue, saturation, and

luminance values of pond color on pond depth and ice thickness, and they are determined by normalizing the square of correlation coefficient $R^2$ between the HSL values and the measured Hi and Hp. According to the statistical analyses in Istomina et al. (2016), there is $c_H$ = 0.255, $c_S$ = 0.712, and $c_L$ = 0.033 (the Table to calculate these values was included in the revised manuscript).

[Figure]

Figure B. This is a subset of ice-thickness retrievals for $H_i$ < 1 m. R is the correlation coefficient between simulated and measured $H_i$. P is the significance level of the correlation. ε is the root-mean-square error, and <ξ> is the mean of relative error in simulated $H_i$.

The different sensitivity of hue, saturation, and luminance values of pond color on $H_i$ and $H_p$ were considered using the parameters $c_H$, $c_S$, and $c_L$ in Eq. (7). Then the results of ice thickness retrievals were improved. Especially for thin ice $H_i$ < 1 m (Figure B), the correlation coefficient between simulated and measured ice thickness $R^2$ = 0.671 and the correlation is significant (P = 0.02). The relative error ξ between simulated and measured values presents an average of 29%.

We think the result is acceptable considering available data is very limited. More validations from field observations in future are needed in order to improve the retrieve model and reduce the errors.

References

Eicken, H., Alexandrov, V., Gradinger, R., Ilyin, G., Ivanov, B., Luchetta, A., Martin, T., Olsson, K., Reimnitz, E., Pac, R., Poniz, P. and Weissenberger, J. 1994. Distribution, structure and hydrography of surface melt puddles, Ber. Polarforsch, 149, 73–76.

Flocco, D., D.L. Feltham, E. Bailey, and D. Schroeder. 2015. The refreezing of melt ponds on Arctic sea ice, J. Geophys. Res. Oceans, 120, 647–659, doi:10.1002/2014JC010140.

Grenfell, T.C., and D.K. Perovich. 2008. Incident spectral irradiance in the Arctic Basin during the summer and fall, J. Geophys. Res., 113, D12117, doi:10.1029/2007JD009418.

Istomina, L., Melsheimer, C., Huntemann, M., Nicolaus, M. and Heygster, G. 2016. Retrieval of sea ice thickness during melt season from in situ, airborne and satellite imagery, IEEE International Geoscience and Remote Sensing Symposium (IGARSS), Beijing, 7678–7681, doi: 10.1109/IGARSS.2016.7731002.

Light, B., Maykut, G. A. and Grenfell, T.C. 2004. A temperature-dependent, structural-optical

model of first-year sea ice, J. Geophys. Res., 109, C06013, doi:10.1029/2003JC002164.

Lu, P., M. Leppäranta, B. Cheng, and Z. Li. 2016. Influence of melt-pond depth and ice thickness on Arctic sea-ice albedo and light transmittance, Cold Reg. Sci. Technol., 124, 1–10, doi:10.1016/j.coldregions.2015.12.010.

Perovich, D.K. 1990. Theoretical estimates of light reflection and transmission by spatially complex and temporally varying sea ice covers, J. Geophys. Res., 95(C6), 9557–9567.

Podgorny, I.A., and T.C. Grenfell. 1996a. Partitioning of solar energy in melt ponds from measurements of pond albedo and depth, J. Geophys. Res., 101(C10), 22737–22748.

Podgorny, I.A., and T.C. Grenfell. 1996b. Absorption of solar energy in a cryoconite hole, Geophys. Res. Lett., 23, 2465–2468.

Skyllingstad, E.D., and C.A. Paulson. 2007. A numerical study of melt ponds, J. Geophys. Res., 112, C08015, doi:10.1029/2006JC003729.

Skyllingstad, E.D., C.A. Paulson, and D.K. Perovich. 2009. Simulation of melt pond evolution on level ice, J. Geophys. Res., 114, C12019, doi:10.1029/2009JC005363.

Smith, R.C., and K.S. Baker. 1981. Optical properties of the clearest natural waters (200–800 nm). Applied Optics, 20: 177–184.

Taylor, P.D., and D.L. Feltham. 2004. A model of melt pond evolution on sea ice, J. Geophys. Res., 109, C12007, doi:10.1029/2004JC002361.

Tsamados, M., Feltham, D., Petty, A., Schroeder, D. and Flocco, D. 2015. Processes controlling surface, bottom and lateral melt of Arctic sea ice in a state of the art sea ice model, Phil. Trans. R. Soc. A, 373, 20140167, doi:/10.1098/rsta.2014.0167.

---

## Author Comment (AC3) · 20 Dec 2017

**J. Hutchings (Editor)

The authors are grateful to the editor for the constructive comments. These comments have been helpful to improve the manuscript a lot. Our responses to the comments are addressed point by point.

Dear Peng Lu and co-authors, Thank you for your contribution. I am interested in receiving your response to the reviews. I have provided some additional comments below. At this stage I have not checked the paper for continuity. I do note that the paper is well written and English clear (thank you), and as I expect you will make some substantial revisions to the paper I am holding off on a through proof-read until after revision.

Please consider another paper in the Cryosphere Discussion that is on the topic of reflectance of melt ponds. I would be very interested in your opinion on the complimentary nature of your work to this. You can find the paper at https://www.the-cryospherediscuss.net/tc-2017-150/tc-2017-150.pdf, or I can send you a pdf if you need.

Reply: Thanks for your promotion. Actually Larysa Istomina and Georg Heygster in that paper Malinka et al. (2017), are also co-authors of this paper.

In Malinka et al. (2017), a RTM for melt ponds different to ours is developed based on Makshtas and Podgorny (1996), and the melt-pond reflectance was estimate by using their RTM, and pond depth and ice thickness were also retrieved using measured spectral albedo. The latter part of Malinka et al. (2017) has the same focus with the discussions in section 4.2 of our paper, but the models and the parameters employed to retrieve are different with each other. We also now cited their results in the conclusion part as "A recent publication by Malinka et al. (2017) suggested another way to determine pond depth and ice thickness from measured spectral albedo of melt ponds. They obtained better retrievals of Hi and Hp partly because they used more complicated spectra as input compared with our case.".

We think these two papers not only prove that problems on melt ponds are really focus of scientists, but also can promote the improvements in the scientific field through academic debate.

In general, please check that you are citing the original source of information. Was Polashenski and Perovich (2012), line 15, page 2, the original source of the 7 stage model for albedo evolution in summer? I recall Hajo Eicken and Don Perovich talking about this much earlier.

Reply: We checked the paper of Perovich and Polashenski (2012). The seven-phase evolution is a main finding of that paper, and was also clearly stated in the abstract section of Perovich and Polashenski (2012). So it should be the original source.

I am curious, could your model be extended to clear skies with non-diffuse illumination? Would this allow you to identify the thickness of ice under melt ponds from satellite imagery such as provided by MODIS? Is this inverse problem one you considered? How much influence does assuming overcast skies have on the comparison with in-situ observations? Did you only consider the sub-set of overcast data in the comparison, or does this include data for all skies?

Reply: (1) A diffuse incident solar radiation is the basic assumption of the present radiative transfer model, so non-diffuse illumination under clear skies cannot be investigated in this study. It is the same with the studies of Perovich (1990), Taylor and Feltham (2004), Flocco et al. (2015) who employed the similar two-stream radiative transfer model for sea ice or melt pond. Besides, overcast sky is prevailing although not always during summer in Arctic. It is acceptable if most situations can be treated in this paper.

(2) The spatial scale of melt ponds is small as comparing with the resolution of satellite instruments such as MODIS. So we think it is very difficult to observe pond color by satellite remote sensing, as one of the reviewers said. But hand-held photography, shipboard photography, and aerial photography are very effective ways to get the small-scale information on ice surface. Especially, with the wide applications of unmanned aerial vehicles (UAV) in sea ice investigations, it is easy for UAVs equipped with a digital camera to get the information of pond color although within a relative small scale. During Chinese Arctic Expeditions, such kind of equipment has been tested and pictures were obtained. This is exactly what we considered the inverse problem for.

We now added a new paragraph in section 4.2 to clearly state the limitations and applicability of the color-retrieval method, including the ideas presented in (1) and (2).

(3) During in-situ observations, the sky conditions were reported overcast during the optical measurements. It agreed with the assumption in the present model, and the influence of the assumption on the comparison can be ignored. We now add a detailed description on field conditions during the measurements of Istomina et al. (2016).

In the figure captions, I assume "true color" refers to the modeled color of the melt pond. Can you clarify.

Reply: Yes, it is the "simulated color".

I had some difficulty in following your discussion on retrieval of ice thickness from pond color. I feel you need to clarify the discussion as to the parameters that confound the inverse solution. It would help to provide the evidence for this. In particular, the paragraph on lines 10-15, page 12, is vague what the competing parameters in the sky and ice conditions are and how they counteract each other such that it might not be possible to find a single solution based on melt pond color. Given you are justifying the value of your work based on the possibility of developing ice thickness products from satellite and camera observations I feel this needs to be addressed much more carefully in your analysis and discussion.

Reply: We revised the section 4.2 and the revised contents include:

(1) Pond color is a function of pond depth, underlying ice thickness, IOPs of sea ice, and incident solar radiation in the present study. Among them, pond depth and underlying ice thickness are the primary factors according to the sensitivity analyses, and IOPs and incident solar radiation can be assigned with empirical constants for melting sea ice in summer. Then there is (color) = $f(H_i, H_p)$, and the inverse problem we focused on is $(H_i, H_p) = f^{-1}(color)$.

(2) The paragraph on lines 10-15, page 12 is a little confusing because it provided a comparison between Fig. 10 (results of the positive problem) and Fig. 11 (results of the inverse problem). So we removed this paragraph, and focused only on the inverse problem in section 4.2.

(3) The retrieving model to solve the inverse problem was now improved in Eq. (7), and different contributions from the hue, saturation, and luminance values of pond color were considered according to the statistical analyses in Istominia et al. (2016). Then the retrievals of ice thickness were highly improved as comparing with in-situ measurements, especially for thin ice with $H_i$ < 1 m. It also argued for the possibility of our method.

Details can be seen in our reply to the last comment of Referee #2.

(4) We added a new paragraph in section 4.2 to clarify the limitations and applicability of the color-retrieval method. Satellite remote sensing is not the direct application of the method. Instead, UAVs equipped with a digital camera are able to get the information of pond color within a relative small scale, and such equipment has also been tested during Chinese Arctic Expeditions. More validations are of course necessary to improve the robustness of the method, but at least in present, the possibility of the new method is still encouraging.

Please consider acknowledging those who collected the data you use in this study. Looking forward to your response, Jenny

Reply: Yes, we added such acknowledgement: "The authors are grateful to the scientific party of the ARK 27/3 cruise for making the sea ice optical measurements possible. Special thanks are expressed to Marcel Nicolaus for organizing the logistics and to the Sea Ice Physics group on board for assisting with the measurements. Three anonymous reviewers and the editor Jennifer Hutchings are also acknowledged for their constructive comments to highly improve the manuscript".

References:

Flocco, D., D.L. Feltham, E. Bailey, and D. Schroeder. 2015. The refreezing of melt ponds on Arctic sea ice, J. Geophys. Res. Oceans, 120, 647–659, doi:10.1002/2014JC010140.

Istomina, L., Melsheimer, C., Huntemann, M., Nicolaus, M. and Heygster, G. 2016. Retrieval of sea ice thickness during melt season from in situ, airborne and satellite imagery, IEEE International Geoscience and Remote Sensing Symposium (IGARSS), Beijing, 7678–7681, doi: 10.1109/IGARSS.2016.7731002.

Makshtas, A. P. and Podgorny, I. A. 1996. Calculation of melt pond albedos on arctic sea ice, Polar Res., 15 (1), 43-52.

Malinka, A., Zege, E., Istomina, L., Heygster, G., Spreen, G., Perovich, D., and Polashenski, C. 2017. Reflective properties of melt ponds on sea ice, The Cryosphere Discuss., https://doi.org/10.5194/tc-2017-150, in review.

Perovich, D. K. and Polashenski, C. 2012. Albedo evolution of seasonal Arctic sea ice, Geophys. Res. Lett., 39, L08501, doi:10.1029/2012GL051432.

Perovich, D.K. 1990. Theoretical estimates of light reflection and transmission by spatially complex and temporally varying sea ice covers, J. Geophys. Res., 95(C6), 9557–9567.

Taylor, P.D., and D.L. Feltham. 2004. A model of melt pond evolution on sea ice, J. Geophys. Res., 109, C12007, doi:10.1029/2004JC002361.

---

## Author Comment (AC4) · 20 Dec 2017

The authors are grateful to the referee for the constructive comments that helped to improve the manuscript substantially. Our responses to the comments are addressed point by point below.

General Comment: This paper describes investigation of the color of melt pond on the Arctic sea ice simulated by the radiative transfer model and validation using field observations. Such sensing and analyzing melt pond may become increasingly important for detecting progress of warming in the Arctic Ocean. Therefore, I recommend this paper for publication. However, I have a couple of major and minor comments that should be considered.

Major comment 1): In section 3.1, the authors mention the effects of melt depth and underlying ice thickness (P6. L18 – P7. L5). In addition, the effects of albedo and color of melt pond are also considered. The authors describe the pond color depends on underlying ice thickness and the possibility of estimation of ice thickness from the pond color. Basically, the pond color on first-year ice (FYI) containing brine and sea water indicates various gray depending on pond depth. The pond color on multi-year ice (MYI) displays green and blue. Thus the pond color also depends on underlying ice types (FYI or MYI). I recommend to add a description about the effect of ice type difference for same ice thickness. This explanation is expected to make the validity of this manuscript increase.

Reply: We added new descriptions on the effect of ice type on pond color in section 3.1 as "Basically, melt ponds on FYI in Arctic are shallow and flat, resulting in various gray color tones, while MYI melt ponds are always deep and narrow, displaying green and blue (Polashenski et al., 2012; Webster et al., 2015). These agree well with the variations in Fig. 4f".

Major comment 2): I agree the result of the comparisons with field observations described in section 3.5. However, the description of the quantitative measurements for pond color by Istomina et al. (2016) is incomplete. Fog appears frequently during summer and observation of pond color seem to be affected by fog. The authors should mention the influence of fog during summer.

Reply: We agree with reviewer. The fog will give impact on the pond color, especially if one took the photos from a distance, e.g. from helicopter or something like that. In Istomina et al. (2016), "fog indeed happened during the field work, but the hand-held camera was very close to the measured ponds and the work was stopped for heavy fog conditions", so the influence of fog on the obtained pond color was limited in this study.

We have added a detailed description on how the pond color was photographed during field investigations in the revised manuscript.

Major comment 3): According to Fig. 11, a good agreement can be found for thin ice with ice thickness < 1 m (P12. L19-L20). I would like to suggest that the color-retrieval method using a RTM is useful to estimate thin ice thickness because sea ice thickness has been declined in recent years. This is not discussed in a convincing way. In order to understand the argumentations given in the manuscript, I recommend to add discussion about when and

where the color-retrieval method is useful. I think the valid area and period of the color-retrieval method are mainly ice edge and in late-summer, respectively.

Reply: Thank you for this very good comment. A new paragraph was now added to the end of section 4.2 to tell the limitations and applicability of the color-retrieval method. It mainly includes:

(1) This method is valid for thin ice with thickness less than 1 m, and when the melt ponds on top of ice are open or just covered by very thin ice. Frozen melt ponds with a snow or thick ice cover, having an obviously different appearance from open ponds, are excluded from this method.

(2) Overcast sky conditions are preferable for this method. They are prevailing although not always present during summer in Arctic.

(3) It is still difficult for the satellite instruments to detect melt-pond color because of the small spatial scale of melt ponds. In contrast, hand-held photography, ship-borne photography, and airborne photography are very effective ways to get the small-scale information on ice surface and provide a basis for ice thickness retrievals. Especially, with unmanned aerial vehicles (UAVs) equipped with a digital camera it is easy to observe sea ice surface features, including melt-pond color, at a floe scale.

Major comment 4): The manuscript describes that the result shown in Fig. 11 is still encouraging (P13. L1-L5). However, it is difficult to agree a new way of determining the sea-ice thickness. To clarity the validity of the color-retrieval method using RMT, I recommend to redraw plots of the ice thickness less than 1 m and more than 1 m separately in Fig. 11b. Adding the correlation coefficients, significance levels, and root mean square errors in Fig. 11 is also recommended.

Reply: Revised accordingly. We have improved the retrieve model and the results showed some improvement for thin ice thickness detection. A subplot for $H_i < 1$ m was also presented and all necessary statistical parameters were included.

Please check our reply to the last comment of Referee #2 for details.

Minor comments:

1) P2-L1: Studies on melt ponds area more than three aspects. For example, the studies using synthetic aperture radar and passive microwave sensor should be included. There are not many papers about remote sensing of melt pond by satellite. Recently Tanaka et al. (2016) reported estimation of melt pond fraction using satellite microwave radiometer. I recommend to cite their paper in this section. Tanaka, Y., K. Tateyama, T. Kameda, and J. K. Hutchings (2016), Estimation of melt pond fraction over high concentration Arctic sea ice using AMSR-E passive microwave data, J. Geophys. Res. Oceans, 121, doi:10.1002/2016JC011876.

Reply: Thanks for your recommendation. Satellite remote sensing on melt pond has been included in the three aspects we stated on P2. And we now added the new reference there.

2) P3-L14: RTM was investigated the dependence of apparent optical properties (AOPs), particularly albedo and transmittance, on sky conditions, pond depth, ice thickness, and the inherent optical properties (IOPs) of ice and water (Lu et al., 2016). That is worth mentioning as well. For example, it would be essential to show about the broadband albedo were higher

on overcast days than on clear days by 0.01 in August.

Reply: The AOPs of melt ponds have been investigated thoroughly in Lu et al. (2016), and therefore were not the subjective of the present study. The color of melt pond is the focus here rather than the surface albedo.

References

Istomina, L., Melsheimer, C., Huntemann, M., Nicolaus, M. and Heygster, G. 2016. Retrieval of sea ice thickness during melt season from in situ, airborne and satellite imagery, IEEE International Geoscience and Remote Sensing Symposium (IGARSS), Beijing, 7678–7681, doi: 10.1109/IGARSS.2016.7731002.

Polashenski, C., Perovich, D. and Courville, Z.: The mechanisms of sea ice melt pond formation and evolution, J. Geophys. Res., 117, C01001, doi:10.1029/2011JC007231, 2012.

Webster, M. A., Rigor, I. G., Perovich, D. K., Richter-Menge, J. A., Polashenski, C. M. and Light, B.: Seasonal evolution of melt ponds on Arctic sea ice, J. Geophys. Res. Oceans, 120, doi:10.1002/2015JC011030, 2015.

---

## Author Response (AR2)

**Comments to the Author from Editor Jennifer Hutchings**

Thank you very much for your detailed response to the reviewers comments. There are still some concerns that need to be addressed before this paper is suitable for publication in The Cryosphere. Please consider the further comments of reviewer 1. Importantly, the paper could benefit from professional English proof reading.

Reply: The latest major revised manuscript has been checked by a native English speaker.

I would like to see more clarity in your argument that pond color is related to ice thickness beneath the pond. In particular, you have no quantitative discussion on the dependence of this relationship on your assumptions for optical properties of the ice.

Reply: We have given the relationship between pond color vector (r, g, b intensities) and ice thickness ($H_i$) and pond depth ($H_p$) in Fig. 4 (c-e) using the default values of incident solar radiation $F_0$ and ice optical properties.

[Figure]

Figure 4. Variations of melt-pond color with pond depth $H_p$ and underlying ice thickness $H_i$ using default values of $F_0$ and ice optical properties. (c–e) denote intensities of red, green, and blue components scaled in the range of 0–1.

The dependence of the relationship on ice optics is only investigated for a typical case of $H_i$ = 1 m and $H_p$ = 0.3 m using the model (see Fig. 7 in the manuscript). Such dependence can be extended for a large range of $H_i$ and $H_p$ as given below. Now if we change the value of $F_0$ from the default value to another possible value in summer Arctic according to Fig. 5a in the manuscript, we can get the variations in the pond color resulted from varying $F_0$ as comparing with Fig. 4 (c-e). The results are shown in Fig. S1.

[Figure]

Figure S1. Variations in the (a) r, (b) g, (c) b intensities if we only change $F_0$, from August 7 to September 10 (Fig. 5a in the manuscript) as comparing with the default results of pond color shown in Fig. 4 (c-e).

Similarly, if we only change ice scattering coefficient $\sigma_i$, the resulted difference in pond color is shown in Fig. S2. And if ice absorption coefficient $k_{\lambda,i}$ is altered, the difference in pond color is also shown in Fig. S3. In principle, the model works with non-dimensional absorption and scattering parameters $H_i \cdot \sigma_i$ and $H_i \cdot k_{\lambda,i}$ and $H_p \cdot k_{\lambda,w}$ and errors in the absorption and scattering coefficients bring corresponding errors to estimated ice thickness. Since the main uncertainty is in the scattering coefficient and scattering is the governing process in attenuation of radiation in ice, in the first order, the relative error of r % in the scattering coefficient causes relative error of r % to the estimated ice thickness.

[Figure]

Figure S2. Variations in the (a) r, (b) g, (c) b intensities if we only change $\sigma_i$, from 2.5/m to 1.2/m, as comparing with the default results of pond color shown in Figure 4 (c-e).

[Figure]

Figure S3. Variations in the (a) r, (b) g, (c) b intensities if we only change $k_{\lambda,i}$, from max. to min. (Figure 3 in the manuscript) as comparing with the default results of pond color shown in Figure 4 (c-e).

We can see the impact of $F_0$ on the relationship between pond color and $H_i$ and $H_p$ in Fig. S1. As the value of $F_0$ changes from the default value of August 7 to September 10 (lowest level in Fig. 5a), the red intensity decreases by at most 0.07, and the green intensity is nearly constant, and the blue intensity increases by at most 0.2. The variation trend is similar with that in Fig. 5b, and the resulted variation is only significant in the blue intensity for very thick ice ($H_i > 3$ m) as comprising with Fig. 4 (c-e).

In Fig. S2, as ice scattering coefficient $\sigma_i$ changes from 2.5/m to 1.2/m, the red intensity decreases by at most 0.14, and the green and blue intensities decrease by at most 0.15. The resulted difference in pond color is large for thin ice and small for thick ice in Fig. S2. It can

be explained by the impotant role of scattering in ice on the upwelling irradiance from pond surface for thin ice. For thick ice, most irradiance have been dissapated due to absorption in ice, and any further changes in scattering coefficient or ice thickness will not change the upwelling irradiance very much. This has been discussed in Lu et al. (2016).

In Fig. S3, although the absorption coefficient of sea ice changes from its maximum to minimum (cf. Fig. 3 in the mannuscript), the resulted variations in pond color are below 0.08, similar with the results of Fig. 7b. That is, the influence of ice absroption on pond color can be ignored.

These three figures (Figs. S1-S3) gave the impacts of $F_0$ and ice optical properties on the relationship between pond color and $H_i$, $H_p$. And they are results for all possible values of $H_i$ and $H_p$, not for a typical case of $H_i$ = 1 m, and $H_p$ = 0.3 m, whose results are shown in the Fig. 5a, Fig. 7a, and Fig. 7b, respectively in the manuscript.

Figs. S1-S3 are not included in the revised manuscript to avoid any further complexity and difficulties. But the illustrated outcome is explained verbally in the text, and we combined the results of Figs. S1-S3 with the revised Fig. 9 in the manuscript, and we added the errors in simulated pond color due to different selections of the values of $F_0$ and ice optical properties. Please see the reply to the comments on Fig. 9 below.

Abstract: Note the comments from reviewer 1. If pond color is not related to Hi, you can not claim this here. Further in the manuscript you identify that the relationship is only true for ice thinner than 1m. This is important to include in the abstract, if it is indeed true.
Reply: We made our argument clear in the revised manuscript, i.e. the pond color has a certain relationship with the sea ice thickness below melt pond when less than 1 m.

Page 1 line 9: Capitalize 'using' Please check your grammer throughout.
Reply: This sentence is: "Pond color, which creates the visual appearance of melt ponds on Arctic sea ice in summer, is quantitatively investigated using a two-stream radiative transfer model for ponded sea ice.".

Page 1, line 25: " lower the surface albedo from as high as 0.8 (snow) to as low as 0.15" strange grammer. rephrase.
Reply: This sentence is rephrased to: "lower the surface albedo from 0.8 (snow) to 0.15 (pond)".

" As a result, melt ponds are an issue as important and inevitable as the dramatic decay of current Arctic sea ice". I don't understand this. Melt ponds are not the issue. You need to rephrase this sentence, or change it for something that makes sense.
Reply: This sentence is rephrased to: "melt ponds play an important role on the dramatic decay of current Arctic sea ice cover (Flocco et al., 2012)".

page 2, line 28: clarify what are trapped ponds?
Reply: The word "trapped ponds" is replaced by "refrozen ponds".

page 3: " Efforts will also be made to find ways to use the information provided by pond color more effectively because this color contains the optical response of melt ponds and sea ice

to incident solar radiation. " an example of a sentence that needs work. First of all, do you want to use the future tense here, perhaps more specifically state "in this paper" and the sentence is convoluted. It is not easy to understand.

Reply: We rephrased the sentence to: "Efforts are also made to find ways to effectively use the information provided by pond color.".

page 4, line 20: again check grammer, remove "and". Please get some professional help in correcting your English.

Reply: The manuscript has been checked by a native English speaker.

page 5, title for section 2.2. From which spectrum? clarify. Perhaps 'observed spectrum'?

Reply: We revised the title to "Estimation of pond color from simulated upwelling spectrum".

page 5: Illuminance Y, and the Y coordinate can be confused. Please consider using different symbols.

Reply: Illuminance Y is actually the value of a color on the Y coordinate in the XYZ color space defined by CIE. To avoid any confusion, we revised the "XYZ coordinates" to "XYZ tristimulus values".

page 7, line 9: "... melt pond albedo is sensitive to Hi for thin ice, but to Hp for thick ice, ..." This sentence does not make sense, is one property (Hi, Hp) sensitive and the other is not?

Reply: When sea ice thickness was below 1.5 m, melt pond albedo was sensitive to the ice thickness and the sensitivity was not affected by the melt pond depth. When sea ice is getting thicker (> 1.5 m), the melt pond albedo is no more sensitive to the ice thickness. We changed the sentence to "the melt-pond albedo depends mainly on $H_i$ for thin ice ($H_i$ < 1.5 m), and on $H_p$ for thick ice ($H_i$ > 1.5 m)".

page 10, line 15: "The agreement is acceptable". In order to make this statement you need to know the sensitivity of hue, saturation and luminance to sky conditions and ice optical properties. I see that you have numerical experiments considering both optical properties of the ice and solar zenith angle, so I expect you can provide some numbers for the sensitivity so we can believe your statement. It is not possible to determine this sensitivity from the manuscript because you only provide a plot (figure 7) of the sensitivity of pond color to optical properties for 1m ice. I need to be able to see that the sensitivity to ice optical properties does not larger than the variation shown in figure that will allow depth to be estimated to a reasonable tolerance for the result to be meaningful. In figure 7 the range of scattering coefficient results in pond colors that are over a larger range than the observations shown in figure 4. What is a reasonable range for the uncertainty in scattering, and how does this impact the results presented in figure 4. (And the same question for other parameters in the model).

Reply: We revised Fig. 9 and added a new paragraph to tell the uncertainty in pond color due to different values of ice optical properties and incident radiation.

The questions on Figs. 4 and 7 can be addressed by Figs. S1-S3 above, where they are not only for $H_i$ = 1 m but for all possible values of $H_i$ and $H_p$. And in the first order, the system

works with non-dimensional optical absorption and scattering lengths H·k and H·σ so that the relative accuracy is proportional to the relative accuracy of the governing optical parameters.

[Figure]

Figure 9: Comparisons of simulated pond color with in-situ measurements by Istomina et al. (2016) in the HSL color space. Points a, b, and c are special cases discussed in the text. The vertical error bars on the simulated color denote the uncertainties due to variations in the incident solar radiation and ice scattering coefficient different from their default values. R is the correlation coefficient between simulated and measured color. P is the significance level of the correlation. ε is the root-mean-square error, and <ξ> is the mean of relative error in simulated color.

"$H_i$ and $H_p$ are changing variables in the calculation. Uncertainties, among others, are the different in-situ conditions, such as sky conditions and ice optical properties, which need to be specified in the model since these in-situ properties were not measured in Istomina et al. (2016). We therefore carried out sensitivity studies by altering the values of $F_0$, $σ_i$, and $k_{λ,i}$ within reasonable ranges for Arctic sea ice in summer to reveal their impacts on the simulated pond color. The negative error bars in the simulated values in Fig.9 are associated with the scattering in ice as $σ_i$ drops from the default value (2.5 m$^{-1}$) to 1.2 m$^{-1}$ (Fig. 7a). The positive error bars are induced by $F_0$ as it decreases from the representative data in August to low levels in September (Fig. 5a). The influence of ice absorption coefficient on the simulated pond color is very limited (< 0.02), similar with Fig. 5b, and therefore not included in the error bars. It is revealed on Fig.9 that the impact of $σ_i$ on the hue and saturation values is less than 0.05, and that on the luminance is less than 0.14. On the contrary, variation in the luminance value due to $F_0$ is less than 0.04, and that in the hue and saturation is less than 0.15. That is, the maximum uncertainty in the simulated hue, saturation, and luminance values will not exceed 0.2 for different combinations of incident solar radiation and IOPs for summer sea ice. More importantly, these variations still locate almost within the range of ± 2ε, namely the 95% confidence interval (Fig. 9). In other words, these experiments underline the importance of $H_i$ and $H_p$ in determining the color of melt ponds compared with other impact factors."

page 11, line 11: I do not think "Besides" should be hear. The ice scattering and the water scattering are separate from each other in the context of this discussion.

Reply: We revised the sentence to: "It is also assumed here that melt pond water is clean and scattering can be neglected".

page 14, line 10: "The result is acceptable because of the very few available data here". Few data points is rarely a reason for a result to be acceptable! There is correlation between the data sets, I would just state the facts and leave out subjective sentences like this one.

Reply: Removed the sentence accordingly.

page 14, line 13: "The results give support for a possible new way method of determining the sea-ice thickness, especially for melting sea ice," In order for this to be true the method needs to have sufficient fidelity to provide meaningful thickness estimates. i.e. what is the resolution of thickness, for thin ice below 1m thick, that could be resolved with the method? Is this resolution a function of thickness, lighting conditions?

Reply: We understand the point, but the resolution and its variations may be difficult to see, because as we have said in the text that the available data is very few (they are only 5 points in the range $H_i$ < 1m, figure 10b). What we want to do in the paper is just to reveal the relationship between pond color and ice thickness. The relationship is solid at least from the statistics of available data at this moment, and this suggests a possibility of ice thickness retrieve. But a robust method of ice thickness retrieve need more field observations. We have stated at the end of the manuscript that further study, especially the *in-situ* measurements are urgently needed in order to validate our method and eventually make the methodology able to resolve the different ice thickness classification on the basis of pond color.

page 16, line 10-15. Very long sentence. Break up in to a least two sentences.

Reply: The sentence is rephrased to: "In comparison, retrieval of ice thickness from pond color has an obvious advantage over all other methods. Hand-held, ship-borne or airborne photography of melt ponds, especially widespread UAVs equipped with a digital camera, is easy to perform during field campaigns.".

page 16, line 17-18: "As the first insight into the color of melt ponds, we tend to pose a possibility instead of draw a conclusion because of the limited available observations so far". "tend to pose" is vague and makes the reader think you are not clear on the future direction. I recommend you rephrase this sentence.

Reply: We rephrased the sentence to: "The possibility of a color-retrieval method was explored in this study using the limited available observations so far.".

Please ensure your figure numbering is correct in the text in relation to the changed figure numbers? I am not sure figure 9 is referred to correctly. Do you have significance and correlation values for both figures 9 and 10? Are you actually improving correlation by taking the small subset of data with Hi < 1m?

Reply: All figure numbers in the text are checked again.

Yes, Figure 9 is correctly referred in the text.

Yes, the statistical parameters are now included in both Figs. 9 and 10.
Yes, the correlation is improved by taking the subset data with $H_i < 1m$.

Figures are missing panel labels. Please place (a), (b) etc appropriately.
Reply: The panel labels are included in Figures 2, 4, 5, 7, and 10 who have subfigures.

If you need any clarification on these comments, please do not hesitate to contact me to set up a time to talk by phone.
Very best regards,
Jenny

**Comments from Anonymous Referee #1**

The authors have completed sufficient revisions to address the concerns raised in my review of the earlier draft. I have a few minor comments on the current draft:

p. 3 line 23-25. "Albedo sensed by spectral radiometers represents the spectrum upwelling irradiance from the surface,…" Well, no, albedo is the ratio of upwelling to downwelling irradiance, it really tells nothing explicitly about the upwelling irradiance. Also, phrase "spectrum upwelling" makes no sense. It is true that the color of a pond is the response of the human eye, and it is true that the upwelling irradiance does depend on the reflected radiation from the pond surface, and the backscattered radiation from the ice and water below. But, not sure why these things are all in the same sentence. This sentence needs to be rewritten.
Reply: This sentence was revised to: "The color of a melt pond is the response of human eyes to the upwelling irradiance from the surface, which consists of the reflected solar radiation from the pond surface and the backscattering radiation from ice and water below.".

P4 line 20: I don't know the word "illuminant"
Reply: "illuminant" refers to the source of light according to the International Commission on Illumination (usually abbreviated CIE for its French name).

p6 line 17-18: "A value of $\sigma_i$ = 2.5 m $^{-1}$ has been promoted by LU16 for summer Arctic sea ice." Is that to say that the authors intend to hold the scattering coefficient for the ice beneath ponds constant? The reason for stating this scattering coefficient should be explained.
Reply: Yes, the scattering coefficient of sea ice is constant except for the sensitivity study of $\sigma_i$ in section 3.3. The reason of selecting the value is explained now. (1) The values of scattering coefficient for eight types of sea ice and snow have been investigated in Perovich (1990), from 0 for bubble-free ice to 800 m$^{-1}$ for cold dry snow. (2) Sensitivity studies conducted in Lu et al. (2016) have revealed that the value of $\sigma_i$ = 2.5 m$^{-1}$ can produce a more comparable melt-pond albedo with field observations than other value of $\sigma_i$. So a value of $\sigma_i$ = 2.5 m$^{-1}$ is also employed in this study.

p6 line 24-25: Pond depth and ice thickness do affect pond albedo and color, but I think the other, perhaps most important, factor is the characteristic scattering of the ice immediately beneath the pond. The analysis later in the manuscript gets around to this point, but I think that idea should be described here.

Reply: We rephrased the sentence to: "The color of a melt pond changes with different factors such as sky conditions, ice properties, and pond depth (Light et al., 2015; Istomina et al., 2016). We investigate the influence of various factors on pond color in the following sections.".

P7 line 8 -9: "Basically, melt ponds on FYI in Arctic are shallow and flat, resulting in various gray color tones, while MYI melt ponds are always deep and narrow, displaying green and blue…" I think this is as result of different pond depths, yes, different ice thickness, yes, but additionally, of different characteristic scattering properties in the FY and MY ice. Also, ponds on MY ice are not necessarily deep and narrow at the beginning of the summer melt season.

Reply: We changed to "Basically, melt ponds on FYI in Arctic are shallow and flat, resulting in various gray color tones, while melt ponds on MYI may have relative larger depth ranges and more complex geometrical patterns, displaying green and blue."

And in the discussion of ice scattering coefficient in section 3.3, we added "Additionally, MYI in Arctic contains much less brine and more gas bubbles than FYI, then the more scattering in MYI is another possible factor causing the different color of melt ponds on MYI and FYI except for $H_i$ and $H_p$ (Fig. 4f).".

P8 line 12: "…sea ice from melting blue ice…" would be clearer if "…sea ice ranging from melting blue ice…"

Reply: Corrected accordingly.

[revised manuscript text omitted]

**Figure 3: Absorption coefficients of clean seawater, pure bubble-free ice and sea ice in the visible band. The water data are from Smith and Baker (1981). The pure ice data are from Grenfell and Perovich (1981) and Warren (1984). The $k_{\lambda,i}$ value was calculated from $k_{\lambda,i} = v_{pi}k_{\lambda,pi} + v_{bp}k_{\lambda,w}$, based on the volume fractions $v_{pi} \geq 60\%$ and $v_{bp} \leq 20\%$ ($v_{pi} + v_{bp} \leq 100\%$) from field observations of summer**
10  **Arctic sea ice (Huang et al., 2013).**

[Figure]

[Figure]

[Figure]

[Figure]

Figure 4: Variations of melt-pond optics and color with pond depth and underlying ice thickness: (a) integrated pond albedo $\alpha_B$, (b) mean wavelength determined by Eq. (1), (c–e) intensities of red, green, and blue components scaled in the range of 0–1, (f) simulated color of the melt pond in the RBG color space according to the colorimetric method defined by Eqs. (2-6). The sky condition is overcast.

[Figure]

**Figure 5: (a) Typical spectral incident solar irradiances in the Arctic summer under a completely overcast sky according to Grenfell and Perovich (2008), and (b) their influence on melt-pond albedo and the *rgb* intensities of pond color for $H_p = 0.3$ m and $H_i = 1.0$ m. The color bar on top of (b) denotes the simulated color of the melt pond under different sky conditions.**

[Figure]

**Figure 6: Normalized values of incident solar radiation under different sky conditions, defined as the ratio of the spectrum in Fig. 5a to the total energy in the visible band.**

[Figure]

[Figure]

Figure 7: Variation of the *rgb* intensities of pond color and melt-pond albedo with the inherent optical properties of underlying sea ice: (a) scattering coefficient and (b) absorption coefficient for $H_p = 0.3$ m and $H_i = 1.0$ m. Note that $\sigma_i$ within 1.2–2.5 m$^{-1}$ is valid for sea ice under melt ponds, and $\sigma_i = 0$ is presented only as a comparison as an idealized purely absorbing medium. The color bar on top denotes the simulated color of the melt pond under different optical properties of sea ice.

[Figure]

Figure 8: Variations of the *rgb* intensities of pond color and melt-pond albedo during the process of sea-ice melting, assuming $H_i + \delta H_p = 1.3$ m. The color bar on the top denotes the simulated color of the melt pond during ice melting.

[Figure]

**Figure 9: Comparisons of simulated pond color with in-situ measurements by Istomina et al. (2016) in the HSL color space. Points a, b, and c are special cases discussed in the text. The vertical error bars on the simulated color denote the uncertainties due to variations in the incident solar radiation and ice scattering coefficient different from their default values. *R* is the correlation coefficient between simulated and measured color. *P* is the significance level of the correlation. *ε* is the root-mean-square error, and <*ξ*> is the mean of relative error in simulated color.**

[Figure]

**Figure 10: (a) Retrievals of underlying ice thickness and pond depth using measured pond colors in Istomina et al. (2016). (b) is a subset of (a) for $H_i < 1$ m. $R$ is the correlation coefficient between simulated and measured $H_i$. $P$ is the significance level of the correlation. $\varepsilon$ is the root-mean-square error, and $<\xi>$ is the mean of relative error in simulated $H_i$.**

---

## Author Response (AR3)

Editor Decision: Publish subject to minor revisions (review by editor) (13 Mar 2018) by Jennifer Hutchings

Comments to the Author:

Dear Drs. Lu, Lepparanta, Cheng, Li, Istomina and Heygster,

Thank you very much for your detailed and considered response to my comments and the reviewers comments.

I disagree that a variation in pond color of 0.08 due to the range in possible absorption coefficient, can be ignored. This signal is about 50% of the pond color sensitivity to pond depth or ice thickness. So it is significant. Or am I misinterpreting your figure captions for S1-S3 and figure 4? It is not clear to me if you are plotting a relative error in all cases, the absolute error (which is my assumption in the statement above) or the anomaly between each sensitivity experiment and the experiment shown in figure 4. You need to clarify what 'Variations of melt-pond color' is, and how you are comparing between the supplemental figures S1-3 and figure 4 (an anomaly or relative difference?).

Reply:

(1) The description "can be ignored" was not clear enough. What we really meant was: the impact of a varying $k_{\lambda,i}$ (absorption coefficient of sea ice) on the r, g, b intensities is relatively smaller compared with those of $F_0$ (incident solar irradiance) and $\sigma_i$ (scattering coefficient of sea ice). Especially for thin ice ($H_i$ < 1 m), the variations in the r, g, b intensities due to different $k_{\lambda,i}$ is less than 0.02 (figure S3). While the variation in the blue intensity due to different $F_0$ is roughly 0.14 (figure S1), and the variations in the green and blue intensities due to different $\sigma_i$ can approach a maximum of 0.15 (figures S2).

(2) It was not the relative errors. Figures S1–S3 showed the absolute values of the changes in the r, g, b intensities. The whole procedure of producing figures S1–S3 are:

a) The melt-pond color is not a single number but presented by a vector combined of three values, i.e., red intensity, green intensity, and blue intensity. Each value ranges within 0-1. So a certain melt-pond color means one combination of r, g and b intensities.

b) In order to reveal the impact of parameters ($F_0$, $\sigma_i$, and $k_{\lambda,i}$) on each intensity, we first made a standard case using default values of the parameters. Variations of each intensity with $H_p$ (pond depth) and $H_i$ (underlying ice thickness) were presented in figure 4(c-e).

c) We then investigated the impact of $F_0$. If it changed from the default value to another value, we obtained a new series of r, g, and b intensities. Figure S1 shows the absolute difference between the new r, g, b intensities and the values shown in figure 4(c-e).

d) Figures S2–S3, absolute difference between the new r, g, b intensities and the values shown in figure 4(c-e), are illustrated for varying ranges of $\sigma_i$ and $k_{\lambda,i}$, respectively.

e) The default values and varying ranges of F0, $\sigma_i$, and $k_{\lambda,i}$ are prescribed based on in-situ observations. They have been described in section 3.

(3) From the comparisons between figure 4(c-e) and figures S1–S3, we can see that the uncertainties in $F_0$, $\sigma_i$, and $k_{\lambda,i}$ do affect the relationship between pond color and $H_i$, $H_p$, but

the levels of the impacts are difference from each other. As a result, we tested the impact of parameter uncertainty on simulated pond color and compared with field measurements in figure 9, and results revealed a statistically good agreement between simulations and measurements of the melt-pond color although the uncertainties of all parameters have been taken into account.

You are perhaps correct that error in absorption does not change the general relationship for particular ice types, though there may be more impact as the ice properties change when it thins or becomes rotten at the pond base. Thank you for adding bars on figure 9. Thank you for making the effort to demonstrate the sensitivity. I am trusting you that the ranges you choose for the parameters in the sensitivity study are the ranges of uncertainty for these.
Reply: Yes, they are.

Line 26 page 10: changing and variables. These two mean the same thing. You can just say 'variables' in the calculation.
Reply: Revised accordingly.

While you are considering minor revisions, please do take the opportunity for one last proof read to ensure your language is precise.
Reply: We have checked the language and make sure the descriptions are precise.

Best regards,
Jenny

Thank you for yours and other anonymous reviewers' valuable comments that helped to improve the manuscript significantly.

Best regards,
Peng Lu and co-authors

[revised manuscript text omitted]

Figure 3: Absorption coefficients of clean seawater, pure bubble-free ice and sea ice in the visible band. The water data are from Smith and Baker (1981). The pure ice data are from Grenfell and Perovich (1981) and Warren (1984). The $k_{\lambda,i}$ value was calculated from $k_{\lambda,i} = v_{pi}k_{\lambda,pi} + v_{bp}k_{\lambda,w}$, based on the volume fractions $v_{pi} \geq 60\%$ and $v_{bp} \leq 20\%$ ($v_{pi} + v_{bp} \leq 100\%$) from field observations of summer
10  Arctic sea ice (Huang et al., 2013).

[Figure]

[Figure]

**Figure 4: Variations of melt-pond optics and color with pond depth and underlying ice thickness: (a) integrated pond albedo $\alpha_B$, (b) mean wavelength determined by Eq. (1), (c–e) intensities of red, green, and blue components scaled in the range of 0–1, (f) simulated color of the melt pond in the RBG color space according to the colorimetric method defined by Eqs. (2-6). The sky condition is overcast.**

[Figure]

**Figure 5: (a) Typical spectral incident solar irradiances in the Arctic summer under a completely overcast sky according to Grenfell and Perovich (2008), and (b) their influence on melt-pond albedo and the *rgb* intensities of pond color for $H_p = 0.3$ m and $H_i = 1.0$ m. The color bar on top of (b) denotes the simulated color of the melt pond under different sky conditions.**

[Figure]

**Figure 6: Normalized values of incident solar radiation under different sky conditions, defined as the ratio of the spectrum in Fig. 5a to the total energy in the visible band.**

[Figure]

**Figure 7: Variation of the *rgb* intensities of pond color and melt-pond albedo with the inherent optical properties of underlying sea ice: (a) scattering coefficient and (b) absorption coefficient for $H_p = 0.3$ m and $H_i = 1.0$ m. Note that $\sigma_i$ within 1.2–2.5 m$^{-1}$ is valid for sea ice under melt ponds, and $\sigma_i = 0$ is presented only as a comparison as an idealized purely absorbing medium. The color bar on top denotes the simulated color of the melt pond under different optical properties of sea ice.**

[Figure]

[Figure]

**Figure 8: Variations of the *rgb* intensities of pond color and melt-pond albedo during the process of sea-ice melting, assuming $H_i + \delta H_p = 1.3$ m. The color bar on the top denotes the simulated color of the melt pond during ice melting.**

[Figure]

Figure 9: Comparisons of simulated pond color with in-situ measurements by Istomina et al. (2016) in the HSL color space. Points a, b, and c are special cases discussed in the text. The vertical error bars on the simulated color denote the uncertainties due to variations in the incident solar radiation and ice scattering coefficient different from their default values. $R$ is the correlation coefficient between simulated and measured color. $P$ is the significance level of the correlation. $\varepsilon$ is the root-mean-square error, and $<\zeta>$ is the mean of relative error in simulated color.

[Figure]

Figure 10: (a) Retrievals of underlying ice thickness and pond depth using measured pond colors in Istomina et al. (2016). (b) is a subset of (a) for $H_i < 1$ m. $R$ is the correlation coefficient between simulated and measured $H_i$. $P$ is the significance level of the correlation. $\varepsilon$ is the root-mean-square error, and $<\zeta>$ is the mean of relative error in simulated $H_i$.